



# Simulation and Data Assimilation in an Idealized Coupled Atmosphere-Ocean-Sea Ice Floe Model with Cloud Effects

Changhong Mou[1], Samuel N. Stechmann[2], and Nan Chen[2]

[1]Department of Mathematics, Purdue University, West Lafayette, Indiana
[2]Department of Mathematics, University of Wisconsin–Madison, Madison, Wisconsin

**Correspondence:** Nan Chen (chennan@math.wisc.edu)





**Abstract.** Sea ice plays a crucial role in the climate system, particularly in the Marginal Ice Zone (MIZ), a transitional area consisting of fragmented ice between the open ocean and consolidated pack ice. As the MIZ expands, understanding its dynamics becomes essential for predicting climate change impacts. However, the role of clouds in these processes has been largely overlooked. This paper addresses that gap by developing an idealized coupled atmosphere-ocean-ice model incorporating cloud

and precipitation effects, tackling both forward (simulation) and inverse (data assimilation) problems. Sea ice dynamics are modeled using the discrete element method, which simulates floes driven by atmospheric and oceanic forces. The ocean is represented by a two-layer quasi-geostrophic (QG) model, capturing mesoscale eddies and ice-ocean drag. The atmosphere is modeled using a two-layer saturated precipitating QG system, accounting for variable evaporation over sea surfaces and ice. Cloud cover affects radiation, influencing ice melting. The idealized coupled modeling framework allows us to study the

interactions between atmosphere, ocean, and sea ice floes. Specifically, it focuses on how clouds and precipitation affect energy balance, melting, and freezing processes. It also serves as a testbed for data assimilation, which allows the recovery of unobserved floe trajectories and ocean fields in cloud-induced uncertainties. Numerical results show that appropriate reduced-order models help improve data assimilation efficiency with partial observations, allowing the skillful inference of missing floe trajectories and lower atmospheric winds. These results imply the potential of integrating idealized models with data assimilation

to improve our understanding of Arctic dynamics and predictions.

## 1 Introduction

Sea ice is a critical component of our climate system, serving both as a reflective shield that deflects solar radiation and as an insulator that regulates oceanic heat Thomas (2017); Weeks (2010). By controlling the exchange of heat, moisture, and momentum between the ocean and atmosphere, sea ice plays a key role in shaping global weather patterns and broader climate

systems. Therefore, understanding the dynamics of sea ice is essential for improving climate models and making accurate predictions about future climate conditions Bigg (2003); Meier et al. (2014); Gildor and Tziperman (2001); Bhatt et al. (2014); Leppäranta (2011); Maslowski et al. (2012); Thomson et al. (2018); Weeks and Ackley (1986); Weiss (2013).

The Marginal Ice Zone (MIZ), the transitional area between the open ocean and consolidated pack ice, is characterized by fragmented ice floes that interact dynamically with oceanic and atmospheric forces, playing a critical role in energy exchange,

ocean circulation, and climate regulation Dumont (2022); Thomson et al. (2018). As the climate warms, the MIZ is expanding, increasing the fragmentation of ice and intensifying ocean-atmosphere interactions. This growth contributes to shifts in ocean currents, weather patterns, and the overall climate system. The expansion also accelerates the ice-albedo feedback, where more open water absorbs solar radiation, driving further ice melt. Given its growing influence, understanding and modeling the MIZ and its ice floe dynamics is essential for predicting climate change impacts and developing effective adaptation strategies

Manucharyan and Thompson (2017); Strong and Rigor (2013); Timmermans et al. (2018); Squire (2020).

In Earth system models, sea ice is typically represented using continuum frameworks with viscous-plastic rheology Hibler III (1979); Hunke and Dukowicz (1997); Tremblay and Mysak (1997); Toyoda et al. (2019), which effectively captures large-scale dynamics but often has difficulties in accounting for brittle behavior and fine-scale fragmentation. These models work well at




basin scales. However, they lack the resolution needed to simulate individual ice floes and their interactions. In contrast, the

Discrete Element Method (DEM) focuses on individual ice floes in Lagrangian coordinates and allows to provide a more detailed representation of local interactions between ice, ocean, and atmosphere Cundall (1988, 1979); Hart et al. (1988). DEM also reduces computational costs for simulating MIZ by eliminating the need for advective transport schemes required in continuum models, while offering flexible spatial resolution, making it particularly useful for modeling the complex dynamics of the MIZ Lindsay and Stern (2004); Manucharyan and Montemuro (2022); Damsgaard et al. (2018); Bouillon and Rampal

(2015); Rampal et al. (2016); Deng et al. (2024).

One aspect that has not received enough attention in previous studies is the effect of clouds on sea ice dynamics. On the one hand, clouds influence the thermodynamics of the atmosphere-ice system by modulating radiative fluxes and precipitation, which subsequently affect ice melting and growth processes Shine et al. (1984); Huang et al. (2019); Liu et al. (2012); Huang et al. (2017); Morrison et al. (2019); Kay and Gettelman (2009). Understanding these interactions is crucial for accurately pre-

dicting sea ice features under different climate scenarios. Including cloud-sea ice interactions in the modeling framework helps enhance our understanding of MIZ dynamics, particularly for studying the response of the DEM to atmospheric influences. On the other hand, significant challenges appear when observing ice floes in the presence of clouds. In situ measurements are often sparse Brunette et al. (2022); Gabrielski et al. (2015); Hutchings et al. (2012); Itkin et al. (2017); Lei et al. (2020) and have limited spatiotemporal resolution Cámara-Mor et al. (2010); Kwok (2018). As a result, satellite imagery is widely used to

monitor ice floe motion in the MIZ, which is then employed to infer ocean currents Manucharyan et al. (2022); Lopez-Acosta et al. (2019); Chen et al. (2022); Covington et al. (2022). However, ice floes can become obscured in satellite images due to intermittent cloud cover. Understanding the accuracy of recovering floe trajectories and the ocean fields with cloud cover facilitates the study of the MIZ.

This paper works toward filling these gaps by developing an idealized coupled atmosphere-ocean-ice model that incorporates

the effects of clouds and precipitation. This model serves several important purposes. It allows the study of the fundamental physics governing interactions between the atmosphere, ocean, and ice floes, which gives a comprehensive understanding of how these components influence each other. It also provides an idealized modeling framework for analyzing the effect of clouds on ice floes. It specifically examines how cloud cover and precipitation modify the energy balance and influence the processes of ice melting and freezing. Moreover, it functions as a testbed for evaluating the accuracy of inferring missing observations

of ice floe trajectories and the underlying ocean fields in the presence of clouds through data assimilation (DA). The primary goals of this study are thus twofold, addressing both forward (model simulation) and inverse problems (DA): to develop and analyze the idealized coupled atmosphere-ocean-ice model that incorporates the effects of clouds and offers insights into the fundamental interactions within the MIZ and to develop and test an efficient DA scheme using reduced-order models aimed at recovering unobserved variables and fields despite limited and uncertain observations.

The remainder of the paper is organized as follows: Section 2 presents the coupled atmosphere-ocean-ice model. Section 3 discusses the development of cheap surrogate forecast models for studying DA. Section 4 provides numerical simulation results that demonstrate the model dynamics while Section 5 shows the skill of the DA results. The paper is concluded in Section 6.





## 2 The Idealized Coupled Atmosphere-Ocean-Sea Ice Floe System

### 2.1 Overview

To incorporate the effects of clouds and precipitation, we develop a coupled atmosphere-ocean-ice system that provides a understanding of the interactions among these components.

In this framework, sea ice dynamics are modeled using the DEM, which represents individual ice floes as circular elements with specific sizes and masses Cundall (1988); Hart et al. (1988). The DEM effectively captures the interactions and shape-preserving behaviors of floes, with their motion driven by atmospheric and oceanic forces. These forces are quantified through

surface integrals over the floes.

Ocean dynamics are modeled using a two-layer quasi-geostrophic (QG) model, known as the Phillips model Vallis (2017); Salmon (1998). This model effectively simulates eddies resulting from baroclinic instabilities, which are crucial for accurately representing oceanic conditions. The model is configured to reflect an Arctic Ocean regime Qi and Majda (2016), capturing key interactions between the ocean and the ice floes.

The atmospheric component employs a two-layer saturated precipitating quasi-geostrophic (PQG) model Smith and Stech-mann (2017); Edwards et al. (2020a, b); Hu et al. (2021). This model addresses the atmospheric forces acting on the ice floes and incorporates the precipitation dynamics that contribute to their growth. The source of water vapor is represented by a parameter quantifying evaporation within the saturated PQG framework Edwards et al. (2020a, b), varying between the sea surface and the ice floe surface. This variation allows the model to simulate how changes in ice floe distribution inversely im-

pact atmospheric thermodynamics. In addition, radiation is incorporated to model the melting of ice floes, with the magnitude adjusted based on cloud thickness within the saturated PQG model.

In the following, Section 2.2 presents the DEM model; Section 2.3 outlines the two-layer QG model; Section 2.4 introduces and discusses the saturated PQG model; and Section 2.5 explores the thermodynamic processes across the different models. Figure 1a illustrates the coupled atmosphere-ice-ocean system along with its interacting components, while Figure 1b provides

a sectional overview of these components within the coupled atmosphere-ice-ocean system, illustrating their interactions.



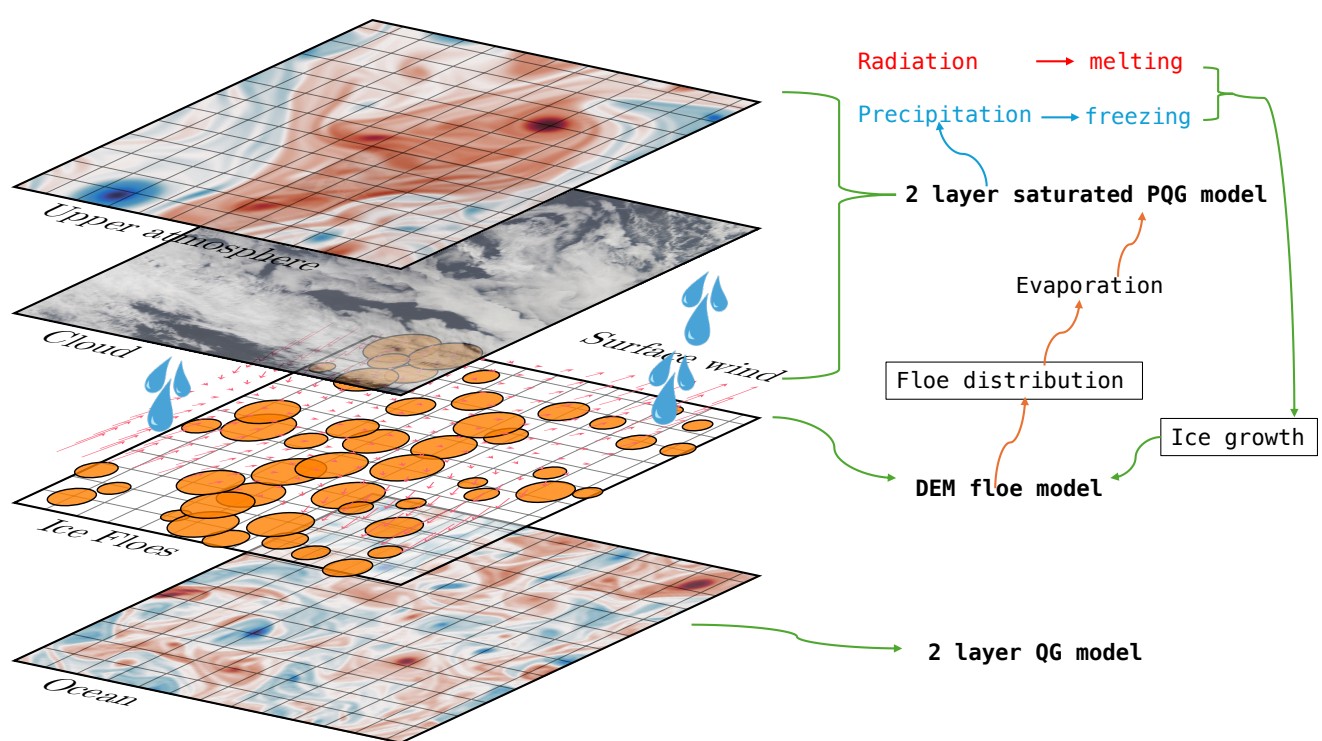

(a) Illustration of the coupled atmosphere-ice-ocean system.

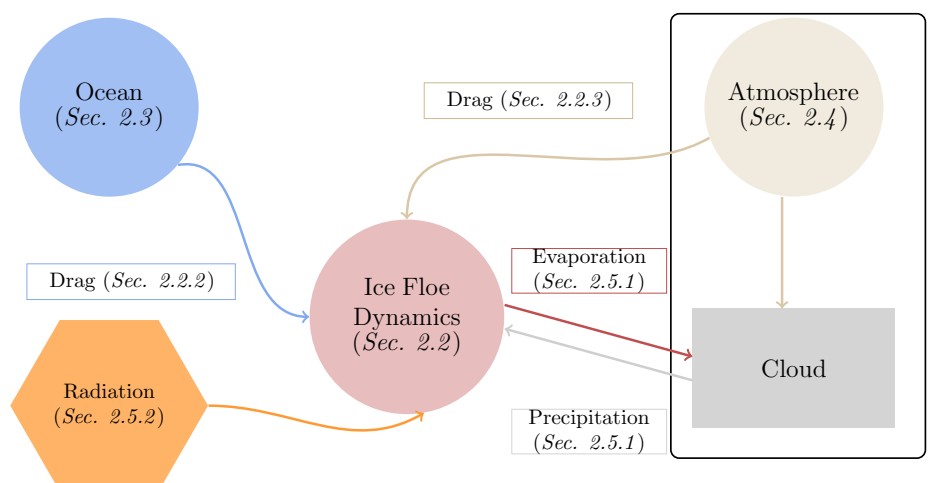

Atmophere-Ocean-Ice Coupling Model, *Sec. 2*

(b) Sectional breakdown of the coupled atmosphere-ice-ocean system.

**Figure 1.** The coupled atmosphere-ice-ocean system.





## 2.2 Ice floe dynamics: the DEM model

### 2.2.1 Governing equations

The DEM is employed to model the motion of sea ice floes, which are simplified as rigid circular bodies Cundall (1979, 1988); Hart et al. (1988); Chen et al. (2021). Each floe is defined by its position, $\mathbf{x}^l = (x^l, y^l)$, and angular displacement, $\Omega^l$, where $l = 1, 2, \ldots, L$ indexes the individual floes. The governing equations for the motion of each floe are as follows:

$$m^l \frac{d^2 \mathbf{x}^l}{dt^2} = \iint\limits_A \mathbf{F}^l \, dA + \mathbf{F}^l_{\text{contact}}, \qquad \text{and} \qquad I^l \frac{d^2 \Omega^l}{dt^2} = \iint\limits_A \tau^l \, dA + \tau^l_{\text{contact}}, \tag{1}$$

where the position $\mathbf{x}^l$ is defined at the center of mass of the $l$-th floe. The second-order time derivative $\frac{d^2 \mathbf{x}^l}{dt^2}$ represents the acceleration of the floe, influenced by the contact force with other floes $\mathbf{F}^l_{\text{contact}}$ and the total external force $\mathbf{F}^l$, integrated over the floe's area $A$. Similarly, the angular acceleration $\frac{d^2 \Omega^l}{dt^2}$ results from the torque due to contacts with other floes $\tau^l_{\text{contact}}$ and the external torque $\tau^l$, also integrated over $A$. Here, $t$ denotes time, $m^l$ is the mass of floe $l$, and $I^l$ is its moment of inertia.

By defining the velocity at the floe's center of mass as $\mathbf{u}^l = (u^l, v^l)$ and the angular velocity as $\omega^l$, the equations in (1) can be reformulated into a set of first-order differential equations:

$$d\mathbf{x}^l = \mathbf{v}^l dt, \tag{2}$$

$$d\mathbf{\Omega}^l = \omega^l dt, \tag{3}$$

$$d\mathbf{v}^l = \frac{1}{m^l} \left( \underbrace{\sum_j \left( \mathbf{f}_{\mathbf{n}}^{lj} + \mathbf{f}_{\mathbf{t}}^{lj} \right)}_{\text{Contact forces}} + \underbrace{\mathcal{D}_o \left( \mathbf{u}_o \left( \mathbf{x}^l \right) - \mathbf{v}^l \right)}_{\text{Ocean drag force}} + \underbrace{\mathcal{D}_a \left( \mathbf{u}_a \left( \mathbf{x}^l \right) - \mathbf{v}^l \right)}_{\text{Atmosphere drag force}} \right) dt, \tag{4}$$

$$d\omega^l = \frac{1}{I^l} \left( \underbrace{\sum_j \left( r^l \mathbf{n}^{lj} \times \mathbf{f}_{\mathbf{t}}^{lj} \right) \cdot \hat{\mathbf{z}}}_{\text{Contact torque}} + \underbrace{\mathcal{T}_o \left( \frac{\nabla \times \mathbf{u}_o \left( \mathbf{x}^l \right)}{2} - \omega^l \hat{\mathbf{z}} \right)}_{\text{Ocean drag torque}} + \underbrace{\mathcal{T}_a \left( \frac{\nabla \times \mathbf{u}_a \left( \mathbf{x}^l \right)}{2} - \omega^l \hat{\mathbf{z}} \right)}_{\text{Atmosphere drag torque}} \right) dt. \tag{5}$$

The notations of variables and parameters in equations (2)–(5) are listed in Table 1. The details of the ocean drag force and torque are covered in Section 2.2.2, while the atmospheric drag force and torque are detailed in Section 2.2.5. The details of the contact forces are provided in Appendix A.

### 2.2.2 Ocean drag force and torque

In this section, we present the mathematical formulations for the drag force and torque exerted on an ice floe by ocean currents. These expressions are crucial for understanding how the ocean's movement affects the floe's velocity and angular velocity.




| $\mathbf{x}^l$ | location | $\mathbf{\Omega}^l$ | toque |
|---|---|---|---|
| $\mathbf{v}^l$ | velocity | $\omega^l$ | angular velocity |
| $m^l$ | mass | $I^l$ | moment of inertial |
| $\mathbf{f_n}^{lj}$ | normal contact force | $\mathbf{f_t}^{lj}$ | tangent contact force |
| $\mathcal{T}_o$ | ocean drag torque function | $\mathcal{T}_a$ | atmosphere drag torque function |
| $\mathcal{D}_o$ | ocean drag force function | $\mathcal{D}_a$ | atmosphere drag foce function |
| $\mathbf{u}_o$ | ocean surface velocity | $\mathbf{u}_a$ | atmosphere surface velocity |

**Table 1.** Notation in DEM model

### 2.2.3 Drag Force.

The velocity of the $l$-th ice floe, denoted as $\mathbf{v}^l$, is affected by the ocean's drag force, which follows a quadratic drag law. The

drag force exerted on the ice floe is expressed as:

$$\mathcal{D}_o^l \left( \mathbf{u}_o \left( \mathbf{x}^l \right) - \mathbf{v}^l \right) = \widetilde{\alpha}(\mathbf{u}_o - \mathbf{v}^l) \left| \mathbf{u}_o - \mathbf{v}^l \right|, \tag{6}$$

where $\mathbf{u}_o(\mathbf{x}^l)$ represents the sea surface current velocity at the centroid of the cylinder floe, i.e., $\mathbf{x}^l$. The coefficient $\widetilde{\alpha}$ is defined as:

$$\widetilde{\alpha} = d_o \rho_o \pi (r^l)^2, \tag{7}$$

where $d_o$ is the ocean drag coefficient, $\rho_o$ is the density of ocean water, and $r^l$ is the radius of the ice floe.

### 2.2.4 Drag torque.

The torque due to ocean drag, denoted as $\tau_o$, influences the angular velocity $\omega^l$ of the ice floe. Assuming a quadratic drag law, the governing equation for the torque is expressed as:

$$\mathcal{T}_o \left( \frac{\nabla \times \mathbf{u}_o}{2} - \omega^l \right) = \widetilde{\beta}_o \left( \frac{\nabla \times \mathbf{u}_o}{2} - \omega^l \right) \left| \frac{\nabla \times \mathbf{u}_o}{2} - \omega^l \right|. \tag{8}$$

Here, $\frac{\nabla \times \mathbf{u}_o}{2}$ represents half of the curl of the ocean surface velocity, corresponding to the angular velocity of the ocean. The coefficient $\widetilde{\beta}_o$ is defined as follows:

$$\widetilde{\beta} = d_o \rho_o \pi (r^l)^4, \tag{9}$$

where $d_o$ is the ocean drag coefficient and $\rho_o$ is the density of the ocean water, consistent with the parameters used in (7).

### 2.2.5 Atmosphere drag force and torque

In this section, we present the mathematical formulations for the drag force and torque exerted on an ice floe by atmospheric wind. These expressions are crucial for understanding how the atmosphere's movement affects the floe's velocity and angular velocity.



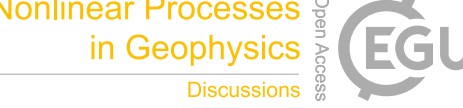

### 2.2.6 Drag force.

The velocity of the ice floe, denoted by $\mathbf{v}^l$, is influenced by the atmospheric drag force, which follows a quadratic drag law.
The equation governing this relationship is given by:

$$\mathcal{D}_a^l \left( \mathbf{u}_a \left( \mathbf{x}^l \right) - \mathbf{v}^l \right) = \widetilde{\alpha} (\mathbf{u}_a - \mathbf{v}^l) \left| \mathbf{u}_a - \mathbf{v}^l \right|, \tag{10}$$

where $\mathbf{u}_a$ represents the atmosphere near-surface wind velocity. The coefficient $\widetilde{\alpha}$ is defined:

$$\widetilde{\alpha}_a = d_a \rho_a \pi (r^l)^2, \tag{11}$$

where $d_a$ is the atmosphere drag coefficient, and $\rho_a$ is the density of the air.

### 2.2.7 Drag torque.

The governing equation for the angular velocity perturbed by the atmosphere, $\mathcal{T}_a$, is a quadratic function, which yields:

$$\mathcal{T}_a \left( \frac{\nabla \times \mathbf{u}_a}{2} - \omega^l \right) = \widetilde{\beta} \left( \frac{\nabla \times \mathbf{u}_a}{2} - \omega^l \right) \left| \frac{\nabla \times \mathbf{u}_a}{2} - \omega^l \right|. \tag{12}$$

where $\widetilde{\beta}_a$ is the coefficient of drag, defined as follows:

$$\widetilde{\beta}_a = d_a \rho_a \pi (r^l)^4. \tag{13}$$

Here, $d_a$ is the atmosphere drag coefficient and $\rho_a$ is the density of the air, consistent with the parameters used in (11).

### 2.3 Ocean dynamics: a two-layer quasi-geostrophic model

The two-layer quasi-geostrophic (QG) model operates in a rotating reference frame, with two layers of equal depth bounded by a rigid lid at the top and a flat bottom. The governing equations of this model are expressed in terms of barotropic and baroclinic modes for potential vorticity (PV) anomalies, with periodic boundary conditions imposed in both the $x$ and $y$ directions Qi and
Majda (2016); Vallis (2017); Salmon (1998). The model is governed by the following equations:

$$\frac{\partial q_1^o}{\partial t} + J(\psi_1^o, q_1) + \beta \frac{\partial \psi_1^o}{\partial x} + U_o \frac{\partial}{\partial x} \left( \Delta \psi_1^o + (k_d^o)^2 \psi_1^o \right) = -\nu_o \Delta^4 q_1^o, \tag{14}$$

$$\frac{\partial q_2^o}{\partial t} + J(\psi_2^o, q_2) + \beta \frac{\partial \psi_2^o}{\partial x} - U_o \frac{\partial}{\partial x} \left( \Delta \psi_2^o + (k_d^o)^2 \psi_2^o \right) = \kappa_o \Delta \psi_2^o - \nu_o \Delta^4 q_2^o, \tag{15}$$

where the subscript $(\cdot)_i^o, i = 1, 2$, denotes the oceanic layers. The term $q_i^o$ denotes the PV and $\psi_i^o$ denotes the streamfunction. The term $k_d^o$ represents the baroclinic deformation wavenumber corresponding to the Rossby radius of deformation $L_d$.
Additionally, $J(A, B) = A_x B_y - A_y B_x$ denotes the Jacobian operator. A large-scale vertical shear $(U_o, -U_o)$, with the same strength but opposite directions, is assumed in the background to induce baroclinic instability. In the dissipation terms on the right-hand sides of the equations, besides hyperviscosity $\nu_o \Delta^4 q_i^o$, only Ekman friction $\kappa_o \Delta \psi_2^o$ is used, with $\kappa_o$ indicating the





strength of the friction applied to the surface layer of the ocean flow. Additionally, the relationships between the PV and the streamfunction are described by the following equations:

$$q_1^o = \Delta\psi_1^o - \frac{(k_d^o)^2}{2}(\psi_1^o - \psi_2^o), \tag{16}$$

$$q_2^o = \Delta\psi_2^o + \frac{(k_d^o)^2}{2}(\psi_1^o - \psi_2^o). \tag{17}$$

### 2.4 Atmosphere dynamics: a two-layer saturated precipitating quasi-geostrophic model

The precipitating quasi-geostrophic (PQG) model, as detailed in Smith and Stechmann (2017); Edwards et al. (2020b); Hu et al. (2021), captures synoptic-scale dynamics, particularly in extratropical regions. This model extends traditional quasi-geostrophic dynamics by incorporating moisture-related thermodynamic processes, such as water vapor dynamics, cloud formation, phase transitions, and precipitation. This integration provides a more realistic representation of the large-scale meteorological patterns commonly observed in these areas.

In this study, we focus on the structure and statistics of total water within fully saturated domains. By employing a fully saturated or convective setup, we simplify the analysis and simulation, avoiding the complexities introduced by phase changes or convective thresholds that could complicate the coupling model. The two-layer fully saturated PQG model yields the following equations:

$$\frac{\partial q_1^a}{\partial t} + J(\psi_1^a, q_1^a) + \beta\frac{\partial \psi_1^a}{\partial x} + U_a\frac{\partial}{\partial x}\left(\Delta\psi_1^a + (k_d^a)^2\psi_1^a\right) = \quad -\nu_a\Delta^4 q_1^a, \tag{18}$$

$$\frac{\partial q_2^a}{\partial t} + J(\psi_2^a, q_2^a) + \beta\frac{\partial \psi_2^a}{\partial x} - U_a\frac{\partial}{\partial x}\left(\Delta\psi_2^a + (k_d^a)^2\psi_2^a\right) = \quad \kappa_a\Delta\psi_2^a - \nu_a\Delta^4 q_2^a, \tag{19}$$

$$\frac{\partial M}{\partial t} + J(\psi_m, M) + v_m\frac{\partial M_{bg}}{\partial y} = -\frac{V_p}{\Delta z}q_{t,m} + E - \nu_a\Delta^4 M. \tag{20}$$

In this model, the subscript $(\cdot)_i^a, i = 1, 2$, denotes the atmospheric layers. The term $q_i^a$ represents the PV, $\psi_i^a$ the streamfunction, and $M$ the balanced moisture variable. Besides, $V_p$ is the precipitation fall speed and $E$ is the evaporation rate. The relationships between vorticity, streamfunction, and PV follow as outlined in equations (16)–(17). The baroclinic deformation wavenumber $k_d^a$ corresponds to the Rossby radius of deformation $L_d$. The total water mixing ratio, $q_{t,m}$, is related to the equivalent potential temperature, $\theta_{e,m}$, as follows:

$$q_{t,m} = M - G_M\theta_{e,m}, \tag{21}$$

where $G_M$ is the ratio of the background vertical gradients of total water mixing ratio and $\theta_{e,m}$, the equivalent potential temperature. The equation for $\theta_{e,m}$ is given by:

$$\theta_{e,m} = \frac{\tilde{L}}{L_{ds}}\frac{\psi_2^a - \psi_1^a}{\Delta z}. \tag{22}$$





The background PV, $PV_{i,bg}$, and balanced moisture, $M_{bg}$, are defined as:

$$PV_{i,bg} = (-1)^i \left(\frac{1}{\Delta z}\right)^2 \left(\frac{\tilde{L}}{L_{ds}}\right)^2 (2U_a y), \tag{23}$$

$$M_{bg} = \frac{1}{\Delta z}\frac{\tilde{L}}{L_{ds}}(2U_a y). \tag{24}$$

where the vertical shear $(U_a, -U_a)$, with the same strength but opposite directions, is assumed in the background to induce baroclinic instability in the atmosphere. Additionally, the characteristic lengthscale $\tilde{L}$ and the saturated deformation lengthscale $L_{ds}$ satisfy the following relationship:

$$(k_d^a)^2 = 8\left(\frac{\tilde{L}}{L_{du}}\right)^2 = \frac{8}{1+G_M}\left(\frac{\tilde{L}}{L_{ds}}\right)^2. \tag{25}$$

### 2.5 Thermodynamic interactions between atmosphere, ocean, and sea ice components

#### 2.5.1 Floe freezing and melting

In the atmospheric model, the magnitude of precipitation is primarily regulated by the variable $q_t$, which is derived from the balanced moisture variable $M$ and the stream functions $\psi_1$ and $\psi_2$. The evolution equation (20) governs the transport of moisture, as well as the falling of precipitation, represented by $-\frac{V_p}{\Delta z}q_{t,m}$, and evaporation, denoted by $E$.

In the initial coupling between ice floes and the atmosphere, several factors are taken into account. First, the precipitation fall speed $V_p$ is assumed to be constant, with the amount of precipitation at each location determined by the variable $q_{t,m}$. Secondly, the evaporation rate $E$ varies between oceanic and ice surfaces Omstedt et al. (1997); Bintanja and Selten (2014). Specifically, evaporation is described by:

$$E(x,y) = \begin{cases} E_i & \text{if } (x,y) \in \mathbb{Q}, \\ E_o & \text{if } (x,y) \in \mathbb{R}^2 \setminus \mathbb{Q}, \end{cases} \tag{26}$$

where $\mathbb{Q}$ represents regions covered by ice floes, and $\mathbb{R}^2$ denotes the double periodic domain of the ocean. In the ice-atmosphere coupled model, the total water content $q_t$ is significantly influenced by spatial variations in the evaporation rate across ice-covered and open ocean regions. Over the open ocean, evaporation typically occurs at a higher and more consistent rate due to the relatively warm surface temperature and the absence of insulating ice cover. This facilitates a steady exchange of moisture between the ocean surface and the atmosphere. In contrast, the evaporation rate over sea ice is markedly lower, as ice is a barrier that reduces direct contact between the water surface and the atmosphere. This difference in evaporation rates is further complicated by the presence of ice floes, which introduce localized variations in the evaporation rate. Specifically, larger ice floes can significantly suppress evaporation in their vicinity, while smaller floes may have a negligible impact. Consequently, the model is required to account for the heterogeneous distribution of evaporation rates to accurately simulate total water content and its dynamics within the coupled sea ice-atmosphere system. This difference is crucial for understanding





the overall moisture balance, cloud formation, and precipitation patterns in polar regions. In particular, the evaporation rate can be parameterized as a function of floe size (radius $r_l$) and the distance from the center of the floe:

$$E(x,y) = E_o - \sum_{l=1}^{L} \tilde{E}_{\text{ice}}(x,y;x_l,y_l,r_l), \tag{27}$$

where

$$\tilde{E}_{\text{ice}}(x,y;x_l,y_l,r_l) = (r_l - r_{thr})^+ \cdot E_{\text{ice}}(x,y;x_l,y_l,r_l). \tag{28}$$

The influence of the $l$-th ice floe on the evaporation rate at a location $(x,y)$ is modeled as

$$E_{\text{ice}}(x,y;x_l,y_l,r_l) = a_l \cdot \frac{1}{\sqrt{2\pi r_l^2}} \exp\left(-\frac{(x-x_l)^2 + (y-y_l)^2}{2r_l^2}\right), \tag{29}$$

where $a_l$ is a scaling parameter given by $a_l = \frac{1}{L_{\text{domain}}} \frac{r_l - r_{thr}}{r_{thr}}$. Here, $r_l$ is the radius of the $l$-th ice floe, and $(x_l,y_l)$ denotes its location. The standard deviation of the Gaussian distribution, representing the spatial influence of the ice floe, is $\sigma(r_l) = r_l$. To

account for the negligible impact of small ice floes, we introduce the threshold $r_{thr}$ in the ramp function $(r_l - r_{thr})^+$, defined as:

$$(r_l - r_{thr})^+ = \begin{cases} r_l - r_{thr} & \text{if } r_l > r_{thr}, \\ 0 & \text{if } r_l \leq r_{thr}, \end{cases} \tag{30}$$

where $r_{thr} = 20\,\text{km}$ is the chosen threshold radius. Ice floes with a radius smaller than $r_{thr}$ are assumed to have a negligible impact on the evaporation rate distribution. The term $(r_l - r_{thr})^+$ in equation (28) ensures that only the portion of the ice floe's

radius exceeding this threshold contributes to the reduction in evaporation. As a result, the overall evaporation rate $E(x,y)$ is determined by subtracting the cumulative effect of all significant ice floes from the constant base evaporation rate over the ocean. The influence of each ice floe is represented by a Gaussian function centered at its location, with its effect scaled by the floe's radius beyond the threshold.

Additionally, precipitation/snow that falls onto the surface of an ice floe contributes to an increase in its height, depth, or

mass Massom et al. (2001); Provost et al. (2017). This increase is uniformly distributed across each floe. The rate of increase in the depth of the floe $l$ over time $\Delta t$ is given by:

$$\frac{dh^l}{dt} = \frac{1}{\rho_w \pi r_j^2} \iint_{\Omega_j} \frac{V_p}{\Delta z} q_{t,m}(x,y) dx dy, \tag{31}$$

where $\Omega_l$ is the subdomain of the $l$-th ice floe and $\rho_w$ is the density of water. It is important to note that with the assumption of ice floes being circular in shape, the mass $m^l$ of each floe can be calculated using the formula:

$$m^l = \rho_{ice} \pi (r^l)^2 h^l, \tag{32}$$

where $\rho_{ice}$ represents the density of ice, and $r^l$ denotes the radius of the floe.





### 2.5.2 Radiation and transfer of radiant energy

The term "insolation" refers to the amount of incoming solar energy Berger (1978). To calculate the energy absorbed by sea ice from sunlight, the intercepted energy is multiplied by one minus the albedo value Miller et al. (2010); Wang et al. (2016).

The albedo quantifies the fraction of light reflected away from the ice, so one minus the albedo represents the fraction of light energy absorbed. The total energy absorbed by the $l$-th sea ice over time $\Delta t$ is given by:

$$E_l^{ice} = \Delta t \iint_{\Omega_l} \gamma(\mathbf{x}) E_s (1 - \alpha) \, dx \, dy, \tag{33}$$

where $E_s$ is known as the solar insolation or solar constant, and $\alpha$ is the albedo of the ice. The variable $\gamma$ represents the fraction of radiation that penetrates through the cloud layer, which varies inversely with the total water content; more total water results

in a lower $\gamma$ value.

To assess how radiation impacts the size of each ice floe, we assume that any change in the ice floe's volume is uniformly distributed across the floe. The formula for the reduction in the depth of the $l$-th floe over time $\Delta t$ can be expressed as:

$$\frac{dh^l}{dt} = -\frac{1}{\rho_{ice} \pi (r^l)^2} \frac{E_l^{ice}}{C_{ice}}, \tag{34}$$

where $C_{ice}$ represents the specific heat capacity of the ice.

## 3 Data Assimilation (DA) of the Coupled System


The coupled model developed above can serve as a testbed for evaluating the accuracy of inferring missing observations of ice floe trajectories and the underlying ocean fields in the presence of clouds through data assimilation (DA). This topic is crucial not only for understanding dynamical coupling and inference capabilities but also as a prerequisite for effective forecasting.

In the presence of cloud cover, DA encounters the challenge of missing observations, raising the question: *"How can we*
*recover unobserved variables and fields when observations are absent?"* However, due to the complexity of the coupled model, using it directly as the forecast model in ensemble DA proves computationally expensive. To address this issue, we develop a low-cost surrogate model for nonlinear DA with partial observations. The surrogate model simplifies the DEM system by replacing contact forces with white noise and utilizing reduced-order models for the atmosphere and ocean in spectral space Chen (2023); Majda and Harlim (2012); Majda et al. (2019); Chen et al. (2022). Such a strategy significantly
reduces computational costs while preserving the essential features of the original model.

Despite the lack of adequate polar observations, satellite data on ice floes and wind conditions can assist in recovering unobserved fields Haugen (2014). We employ the local ensemble transform Kalman filter ensemble (LETKF) Hunt et al. (2007); Bishop et al. (2001) to integrate model forecasts and partial observations through Bayesian updating Evensen (2003); Houtekamer and Mitchell (2005).





### 3.1 Cheap surrogate forecast models for the coupled system

DA involves two key steps: forecasting and analysis. The forecasting step utilizes a forecast model to obtain predicted statistics, which does not necessarily need to be the true underlying dynamics. In practice, the actual system is never fully known, and comprehensive models can be costly to run, particularly for ensemble forecasts. Consequently, using appropriate, inexpensive surrogate or reduced-order models (ROMs) that capture the essential dynamical and statistical features of the original system is often crucial for facilitating practical DA Majda (2016); Farrell and Ioannou (1993); Berner et al. (2017); Branicki et al. (2018); Majda and Chen (2018); Li and Stechmann (2020); Harlim and Majda (2008); Chen and Majda (2018); Kang and Harlim (2012).

In the coupled atmosphere-ocean-sea ice model developed above, the most computationally intensive components, such as detailed floe-floe and ocean-atmosphere interactions, are replaced with suitable surrogates that approximate their behavior with significantly lower complexity. The challenging task of modeling contact forces among sea ice floes is simplified by representing these forces as white noise. By alleviating the computational burden, ROMs enable more frequent updates to the model state using new observational data, thereby enhancing the accuracy and reliability of predictions.

#### 3.1.1 Surrogate forecast model for sea ice motions

The surrogate forecast model utilized in DA starts with a simplified DEM model for the sea ice motions:

$$d\mathbf{x}^l = \mathbf{v}^l dt, \tag{35}$$

$$d\mathbf{v}^l = \frac{1}{m^l} \left( \underbrace{\mathcal{D}_o^l \left(\mathbf{u}_o - \mathbf{v}^l\right)}_{\text{Ocean drag force}} + \underbrace{\mathcal{D}_a^l \left(\mathbf{u}_a - \mathbf{v}^l\right)}_{\text{Atmosphere drag force}} \right) dt + \sigma_v d\mathbf{W}_v^l(t), \tag{36}$$

where $\mathbf{x}^l$ is the position of the ice floe, $\mathbf{v}^l$ is the velocity of the ice floe contributed by the ocean drag force $\mathcal{D}_o^l(\mathbf{u}_o - \mathbf{v}^l)$ and the atmospheric drag force $\mathcal{D}_a^l(\mathbf{u}_a - \mathbf{v}^l)$, $m^l$ is the mass of the ice floe, $\sigma_v$ represents the intensity of the stochastic term, and $\mathbf{W}_v^l(t)$ is a Wiener process representing the stochastic component. The stochastic forcing is used to effectively approximate the instantaneous floe-floe interactions in the forecast system, significantly reducing the computational cost. Equation (35) describes the change in position over time, while Equation (36) describes the change in velocity, incorporating both ocean and atmospheric drag forces, as well as a stochastic term to account for random contact forces.

#### 3.1.2 Surrogate forecast model for the atmosphere dynamics

Recall that the atmospheric wind fields are modeled using the two-layer saturated PQG framework. However, directly applying this two-layer PQG model in DA proves to be computationally expensive. To overcome this challenge, we propose constructing stochastic surrogate models for a limited number of spectral modes of the streamfunction. Given that the upper atmospheric variable is observed, it is essential for the surrogate models to maintain the connection between the upper and lower layers. To this end, we employ both barotropic and baroclinic formulations for the surrogate model that automatically couple the dynamics of the two layers.





The spectral representation of the streamfunctions at two layers is given by:

$$\psi_1 = \sum_{\mathbf{k}} \widehat{\psi}_{1,\mathbf{k}}(t)e^{i\mathbf{k}\cdot\mathbf{x}}, \qquad \text{and} \qquad \psi_2 = \sum_{\mathbf{k}} \widehat{\psi}_{2,\mathbf{k}}(t)e^{i\mathbf{k}\cdot\mathbf{x}}, \tag{37}$$

where $\psi_1$ and $\psi_2$ are the streamfunctions at the first and second layers, respectively. The summation is over wavenumbers $\mathbf{k}$, $\widehat{\psi}_{1,\mathbf{k}}(t)$ and $\widehat{\psi}_{2,\mathbf{k}}(t)$ are the time-dependent spectral coefficients for the respective layers, $i$ is the imaginary unit, and $\mathbf{x}$ represents the spatial coordinates. On the other hand, the barotropic and baroclinic formulations for each Fourier wavenumber yield the following:

$$\widehat{\psi}_{bt,\mathbf{k}} = \frac{\widehat{\psi}_{1,\mathbf{k}} + \widehat{\psi}_{2,\mathbf{k}}}{2}, \qquad \text{and} \qquad \widehat{\psi}_{bc,\mathbf{k}} = \frac{\widehat{\psi}_{1,\mathbf{k}} - \widehat{\psi}_{2,\mathbf{k}}}{2}, \tag{38}$$

where $\widehat{\psi}_{bt,\mathbf{k}}$ and $\widehat{\psi}_{bc,\mathbf{k}}$ are the barotropic and baroclinic streamfunctions for the wavenumber $\mathbf{k}$, respectively. Linear stochastic surrogate models can be constructed for the barotropic and baroclinic streamfunctions for each Fourier wavenumber Chen (2023); Majda and Harlim (2012):

$$d\widehat{\psi}_{bt,\mathbf{k}} = (-\gamma_{bt,\mathbf{k}} + i\omega_{bt,\mathbf{k}})\widehat{\psi}_{bt,\mathbf{k}}dt + f_{bt,\mathbf{k}}dt + \sigma_{bt,\mathbf{k}}dW_{bt,\mathbf{k}}(t), \tag{39}$$

$$d\widehat{\psi}_{bc,\mathbf{k}} = (-\gamma_{bc,\mathbf{k}} + i\omega_{bc,\mathbf{k}})\widehat{\psi}_{bc,\mathbf{k}}dt + f_{bc,\mathbf{k}}dt + \sigma_{bc,\mathbf{k}}dW_{bc,\mathbf{k}}(t), \tag{40}$$

where $\gamma_{bt,\mathbf{k}}, \omega_{bt,\mathbf{k}}, f_{bt,\mathbf{k}}, \sigma_{bt,\mathbf{k}}$ and $\gamma_{bc,\mathbf{k}}, \omega_{bc,\mathbf{k}}, f_{bc,\mathbf{k}}, \sigma_{bc,\mathbf{k}}$ are parameters obtained by calculating the statistical quantities of the time series, while $W_{bt,\mathbf{k}}(t)$ and $W_{bt,\mathbf{k}}(t)$ are independent Brownion motions. In particular, $\gamma_{bt,\mathbf{k}}$ and $\omega_{bt,\mathbf{k}}$ are estimated using the cross-correlation function of the time series of $\widetilde{\psi}_{bt,\mathbf{k}}$:

$$XC_{bt,\mathbf{k}}(t) = \sin(\omega_{bt,\mathbf{k}}t)e^{-\gamma_{bt,\mathbf{k}}t}, \tag{41}$$

where $XC_{bt,\mathbf{k}}(t)$ represents the cross-correlation function for the barotropic component at wavenumber $\mathbf{k}$. The other two parameters can be approximated using the following formulas:

$$T_{bt,\mathbf{k}} = \frac{\gamma_{bt,\mathbf{k}}}{\omega_{bt,\mathbf{k}}^2 + \gamma_{bt,\mathbf{k}}^2}, \quad \theta_{bt,k} = \frac{\omega_{bt,\mathbf{k}}}{\omega_{bt,\mathbf{k}}^2 + \gamma_{bt,\mathbf{k}}^2}, \tag{42}$$

$$f_{bt,\mathbf{k}} = \frac{m_{bt,\mathbf{k}}(T_{bt,\mathbf{k}} - i\theta_{bt,\mathbf{k}})}{T_{bt,\mathbf{k}}^2 + \theta_{bt,\mathbf{k}}^2}, \tag{43}$$

$$\sigma_{bt,\mathbf{k}} = \sqrt{\frac{2E_{bt,\mathbf{k}}T_{bt,\mathbf{k}}}{T_{bt,\mathbf{k}}^2 + \theta_{bt,\mathbf{k}}^2}}, \tag{44}$$

where $m_{bt,\mathbf{k}}$ is the mean, $E_{bt,\mathbf{k}}$ is the variance, $T_{bt,\mathbf{k}}$ and $\theta_{bt,\mathbf{k}}$ are the real and imaginary parts of the decorrelation time, respectively. Concurrently, the streamfunctions in the two layers, $\psi_1$ and $\psi_2$, from the surrogate model can be recovered using the following relations:

$$\widehat{\psi}_{1,\mathbf{k}} = \widehat{\psi}_{bt,\mathbf{k}} + \widehat{\psi}_{bc,\mathbf{k}}, \quad \widehat{\psi}_{2,\mathbf{k}} = \widehat{\psi}_{bt,\mathbf{k}} - \widehat{\psi}_{bc,\mathbf{k}}. \tag{45}$$



The velocities in the two layers of the atmosphere are given by:

$$\mathbf{u}_j^a = (u_j^a, v_j^a) = \left( \frac{\partial \widetilde{\psi}_j}{\partial y}, -\frac{\partial \widetilde{\psi}_j}{\partial x} \right), \tag{46}$$

where $j = 1, 2$ denotes the layer. Here, $\mathbf{u}_j^a$ represents the velocity vector, with $u_j^a$ and $v_j^a$ being the velocity components in the $x$ and $y$ directions, respectively. The surrogate model streamfunction for layer $j$ is given by:

$$\widetilde{\psi}_j = \sum_{\mathbf{k} \in \mathbf{K_r}} \widehat{\psi}_{j,\mathbf{k}}(t) e^{i\mathbf{k} \cdot \mathbf{x}}, \tag{47}$$

where $\mathbf{k}$ denotes the wavenumber vector, $\widehat{\psi}_{j,\mathbf{k}}(t)$ is the time-dependent spectral coefficient for layer $j$, and $\mathbf{K_r}$ represents the set of retained spectral modes.

### 3.1.3 Surrogate forecast model for the ocean dynamics

Recall that the ocean fields are generated by the two-layer QG model. Only the surface layer of the ocean fields is used in coupling the atmosphere and the ice floes. To simplify the model, a stochastic surrogate model is constructed solely for the near surface ocean streamfunction, represented as:

$$d\widehat{\psi}_{o,\mathbf{k}} = (-\gamma_{o,\mathbf{k}} + i\omega_{o,\mathbf{k}})\widehat{\psi}_{o,\mathbf{k}}\, dt + f_{o,\mathbf{k}}\, dt + \sigma_{o,\mathbf{k}}\, dW_{o,\mathbf{k}}(t), \tag{48}$$

where $\widehat{\psi}_{o,\mathbf{k}}$ is the spectral coefficient of the ocean streamfunction at wavenumber $\mathbf{k}$, $\gamma_{o,\mathbf{k}}$ is the damping coefficient, $\omega_{o,\mathbf{k}}$ is the frequency, $f_{o,\mathbf{k}}$ is the forcing term, $\sigma_{o,\mathbf{k}}$ is the noise amplitude, and $W_{o,\mathbf{k}}(t)$ represents the Wiener process.

The parameters $\gamma_o$, $\omega_o$, $f_o$, and $\sigma$ can be approximated similarly using the formulas provided in equations (41) and (44). The velocity of near surface ocean current can be obtained similarly in equations (46) and (47).

### 3.2 Uncertainty in observing ice floes

The presence of clouds poses a significant challenge in observing the location of sea ice floes, especially when using optical and infrared satellite imagery Reiser et al. (2020); Hyun and Kim (2017); Wright and Polashenski (2018). Clouds can obscure the surface, which prevents sensors from capturing clear images of the underneath ice floes. Such an issue causes gaps in observational data, where the exact position and extent of the floes cannot be accurately determined. Note that even when clouds are partially transparent, the scattering and absorption of light can distort observed images, leading to biases in the inferred location and size of the floes. In addition, clouds can form and dissipate relatively rapidly, which complicates the temporal consistency of observations. Consequently, cloud cover introduces a substantial source of uncertainty in observing and monitoring sea ice floes, especially in regions with frequent or persistent cloudiness.

The observability of ice floes varies significantly with their size. Due to their extensive surface area, large ice floes are generally easier to detect in satellite imagery, even in cloudy conditions. These large-size floes may still be partially visible through breaks in the clouds or cloud-affected imagery, allowing for some degree of position and movement tracking. However, the accuracy of this tracking can be affected since cloud-induced distortions may obscure edges or lead to errors in determining





the exact boundaries of floes. In contrast, small ice floes are more susceptible to being completely obscured by clouds. Due to
their limited size, these floes can be entirely covered by clouds, resulting in missing data that complicates the assimilation of
accurate ice dynamics into models. Identifying small floes is also hampered by the limited spatial resolution of many satellite
sensors. Even minor cloud-induced distortions can make small floes undetectable or misclassified, significantly underestimat-
ing their presence and distribution. The difficulty in observing small floes gives a challenge for accurately modeling sea ice
behavior, as these smaller elements can play a critical role in the overall dynamics and thermodynamics of sea ice, particularly
in processes such as melt pond formation and interactions with ocean and atmospheric conditions.

To summarize, while large floes can often be partially observed even in cloudy conditions, the observation of small floes
is much more challenging. This leads to large uncertainties in identifying their location, extent, and influence on the sea ice
system. This disparity emphasizes the need for developing advanced observational techniques and DA methods to bridge the
gaps and reduce the inaccuracies caused by cloud cover.

To represent observational uncertainty in DA, we use the total water content $q_t(\mathbf{x}, t)$ as a controlling factor. Above each floe,
we calculate the mean total water content, $[q_t(\mathbf{x}, t)]$ at time $t$:

$$[q_t(\mathbf{x}, t)] = \frac{1}{|\Omega_l|} \int_{\Omega_l} q_t(\mathbf{x}, t) d\mathbf{x}. \tag{49}$$

This mean value, $[q_t(\mathbf{x}, t)]$, encapsulates the spatial distribution of water content above each ice floe, serving as a proxy for the
uncertainty in observations. Variations in $[q_t(\mathbf{x}, t)]$ from one floe to another can indicate the degree of uncertainty inherent in
the observational data, as it reflects the heterogeneity in the physical characteristics of the ice. In particular, we set a threshold,
$\widetilde{q}_t$, such that the observational uncertainty $\sigma_l^{obs}$ is given by the following:

$$\text{For the } l\text{-th floe,} \begin{cases} \text{If } [q_t(\mathbf{x}, t)] < \widetilde{q}_t & \text{then } \sigma_l^{obs} = 5 \times 10^2 \text{m}, \\ \text{If } [q_t(\mathbf{x}, t)] \geq \widetilde{q}_t & \text{then } \sigma_l^{obs} = 2r_l, \end{cases} \tag{50}$$

where $r_l$ is the radius of $l$-th floe. It is important to note that when the mean total water content over the floe, $[q_t(\mathbf{x}, t)]$, is
high—indicating significant cloud cover—it is still feasible to approximate the location of the floes. However, these estimates
are likely to be substantially inaccurate, particularly when the observational uncertainty, $\sigma_l^{obs}$, is large.

### 3.3   Local ensemble transform Kalman filter

#### 3.3.1   Observational variables

When observing the trajectories of ice floes, a significant challenge arises from cloud coverage, which can obscure these floating
ice bodies in satellite or aerial imagery. To accurately describe the observed trajectories, we start with a general representation
of the trajectories. Let us consider that at time $t$, the trajectory of the $l$-th floe is defined as follows:

$$\mathbf{x}_l(t) = (x_l(t), y_l(t)), \qquad l = 1, 2, \cdots, L, \tag{51}$$





At time $t_k$, when only partial observations of the ice floes are available, the observation of the $l$th floe yields:

$$\mathbf{x}_l^{obs}(t_k) = \left( x_l^{obs}(t_k), y_l^{obs}(t_k) \right), \quad l \in S. \tag{52}$$

Here, $S \subseteq \{1, 2, \cdots, L\}$ represents the subset of observable floes at each $t_k$. Note that not all floes are observable at every $t_k$; only a subset of them is observed. For the $l$th floe, the observation of its trajectory is given by:

$$\mathcal{H}(\mathbf{x}_l)(t_k) = \begin{cases} \mathbf{x}_l^{obs}(t_k), & \text{if there is no cloud;} \\ \text{N/A}, & \text{if there is cloud.} \end{cases} \tag{53}$$

where $\mathcal{H}$ is an observation operator.

To determine whether a given location $(x, y)$ can be observed or not at each observation time $t_k, k = 1, 2, \cdots, N_t$, a simple setting is used based on the total water content $q_t$:

$$\begin{cases} \text{if } q_{t,m}(\mathbf{x}_l^{obs}(t_k)) < q_\epsilon, & \text{then } \mathbf{x}_l^{obs}(t_k) \text{ can be observed,} \\ \text{if } q_{t,m}(\mathbf{x}_l^{obs}(t_k)) \geq q_\epsilon, & \text{then } \mathbf{x}_l^{obs}(t_k) \text{ cannot be observed.} \end{cases} \tag{54}$$

### 3.3.2 Localization

The local ensemble transform Kalman filter (LETKF) Hunt et al. (2007); Bishop et al. (2001) functions similarly to standard ensemble Kalman filters but implements filtering within localized domains. Specifically, the LETKF improves the process by partitioning the global data set into smaller, overlapping regions. Each region is updated independently using local observations. It effectively reduces long-range error correlations and enhances computational efficiency. This localization is essential for large-scale applications. It makes LETKF more scalable and accurate when dealing with spatially-extended systems.

LETKF applies filtering exclusively in physical space for both the trajectories of ice floes and the streamfunctions of the atmosphere and ocean. This unified approach maintains consistency across different types of geophysical data, potentially simplifying the assimilation process and directly addressing spatial correlations.

## 4 Numerical Simulation Results of the Coupled Atmosphere-Ocean-Sea Ice Model

### 4.1 Setup

The domain size considered here is $400 \, \text{km} \times 400 \, \text{km}$ in the marginal ice zone. The size distribution of ice floes follows a power law, represented as $p(r) = ak^a / r^{a+1}$ Stern et al. (2018), where the constants $k$ and $a$ are parameters of the model. The numerical integration time step, $\Delta t$, is set to 58.2 seconds. Three different distributions of floe sizes are considered in this study and each distribution comprises 48 floes, designed to simulate different ice coverage and collision frequency:

- Regime I: Large-size ice floes nearly cover the entire ocean domain, resulting in frequent collisions;

- Regime II: Medium-size ice floes cover approximately half of the ocean domain, leading to less frequent collisions; and



- Regime III: Small-size ice floes occupy only a small portion of the ocean domain, where collisions are rare.

These different scenarios are designed to explore the dynamics of ice floes under different coverage conditions and assess their
impacts on the atmospheric model. The detailed parameters of the DEM model are listed in Table 2.

| Parameter | Value |
| --- | --- |
| Spatial domain, $\Omega$ | $400km \times 400km$ |
| Time Step, $\Delta t$ | 1.2941 hours |
| Density of ice, $\rho_i$ | $10^3\ kg/m^3$ |
| Density of ocean, $\rho_o$ | $1.02 \times 10^3\ kg/m^3$ |
| Density of atmosphere, $\rho_a$ | $1.2\ kg/m^3$ |
| Initial Ice floe height, $h_0$ | $1\,m$ |
| Young's modulus, $E^{lj}$ | $1.2725 \times 10^3\ kg/s^2$ |
| Shear modulus, $G^{lj}$ | $1.3816 \times 10^4\ kg/(m \cdot s)$ |
| Ocean drag coefficient, $d_o$ | $5.5 \times 10^{-3} \times 10^{-4}$ |
| Atmosphere drag coefficient, $d_a$ | $1.6 \times 10^{-3}$ |
| Albedo of ice, $\alpha$ | 0.8 |
| Evaporation rate with ice, $E_i$ | $1.2 \times 10^{-6} kg/(kg \cdot s)$ |
| Evaporation rate without ice, $E_o$ | $2.4 \times 10^{-6} kg/(kg \cdot s)$ |
| Specific heat capacity of the ice, $C_{ice}$ | $334\,KJ/kg$ |

**Table 2.** Parameters in ice floe model.

The model parameters for the two-layer quasi-geostrophic (QG) model are summarized in Table 3. These parameters are specifically tailored to represent conditions typical of the high-latitude ocean regime Qi and Majda (2016).

| Parameter | Value |
| --- | --- |
| Latitude | $72.8°$ |
| Beta value, $\beta$ | $6.74 \times 10^{-12}$ |
| Baroclinic deformation wavenumber, $k_d^o$ | $3.14 \times 10^{-4}$ |
| Background zonal flow, $U_o$ | $6.83 \times 10^{-4} m/s$ |
| Strength of friction $\kappa_o$ | $9.66 \times 10^{-8}$ |
| Hyperviscosity, $\nu_o$ | $6.83 \times 10^{-4}$ |

**Table 3.** Parameters in QG equations for high latitude ocean regime.

The parameters relevant to the atmospheric regime within the framework of the saturated PQG equations are detailed in Table 4 Edwards et al. (2020a, b); Smith and Stechmann (2017); Hu et al. (2021). The atmospheric and oceanic systems exhibit





multiscale characteristics both temporally and spatially. Specifically, typical atmospheric wind velocities range from $8\,m/s$ to $10\,m/s$ (equivalent to $800\,km/day$). They are significantly faster than ocean current speeds, which are around $0.1\,m/s$ (corresponding to $10\,km/day$). In addition, atmospheric winds exhibit rapid temporal changes compared to the relatively slower temporal variations observed in ocean currents.

| Parameter | Value |
|---|---|
| Latitude, | $72.8°$ |
| Beta value, $\beta$ | $6.74 \times 10^{-12}$ |
| Baroclinic deformation wavenumber, $k_d^a$ | $1.26 \times 10^{-4}$ |
| Background zonal flow, $U_a$ | $1.37 \times 10^{-2}\,m/s$ |
| Strength of friction $\kappa_o$ | $2.15 \times 10^{-7}$ |
| Hyperviscosity, $\nu_o$ | $6.83 \times 10^{-4}$ |
| Specific heat, $c_p$ | $10^3\,J\,kg^{-1}\,K^{-1}$ |
| Latent heat factor, $L_v$ | $2.5 \times 10^6\,J$ |
| Background vertical gradient of equivalent potential temperature, $\frac{d\widetilde{\theta}_e}{dz}$ | $1.5\,K\,km^{-1}$ |
| Background vertical gradient of total water, $\frac{d\widetilde{q}_t}{dz}$ | $-0.6 \times 10^{-3}\,kg\,kg^{-1}\,km^{-1}$ |
| Background potential temperature, $\Theta_0$ | $3\,K$ |
| Domain height, $H$ | $10\,km$ |
| Precipitation fall speed, $V_p$ | $2\,m/s$ |
| Solar insolation, $E_s$ | $1,361\,W/m^2$ |

**Table 4.** Parameters in Saturated PQG equations for high latitude atmosphere regime.

## 4.2 Simulated atmospheric and ocean fields

Figure 2 shows the time evolution of the PVs ($PV_1$ and $PV_2$) in the atmosphere from $t = 404.4$ hours to $808.8$ hours and $1213.3$ hours. These snapshots illustrate the development of the PV structures and their time evolution. They indicate the dynamic interactions within the coupled atmosphere-ocean-sea ice system.

**Figure 2.** $PV_1$ and $PV_2$ for atmosphere at different time instances.

Figure 3 shows the upper-layer PV field of the ocean. It reveals distinct patterns in the evolution of the PV structures with both large-scale and small-scale localized features. At time $t = 404.4$ hours, the PV field looks relatively uniform, with moderate gradients indicating steady circulation. When the time arrives at $t = 808.8$ hours, the PV distribution becomes more complicated, with sharper gradients and more eddies. The spatial patterns suggest an increased vorticity intensity with the appearance of turbulent mixing and instabilities in certain regions. At $t = 1213.3$ hours, the PV field is further strengthened. It is more concentrated, indicating a highly dynamic state with stronger interactions among different oceanic layers. The gradient patterns and isolated vortices highlight the nonlinear nature of the response in the ocean to external forcing and the internal energy transfers occurring within the coupled atmosphere-ice-ocean system.

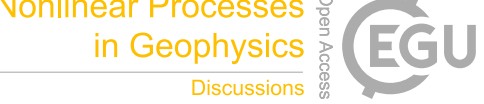

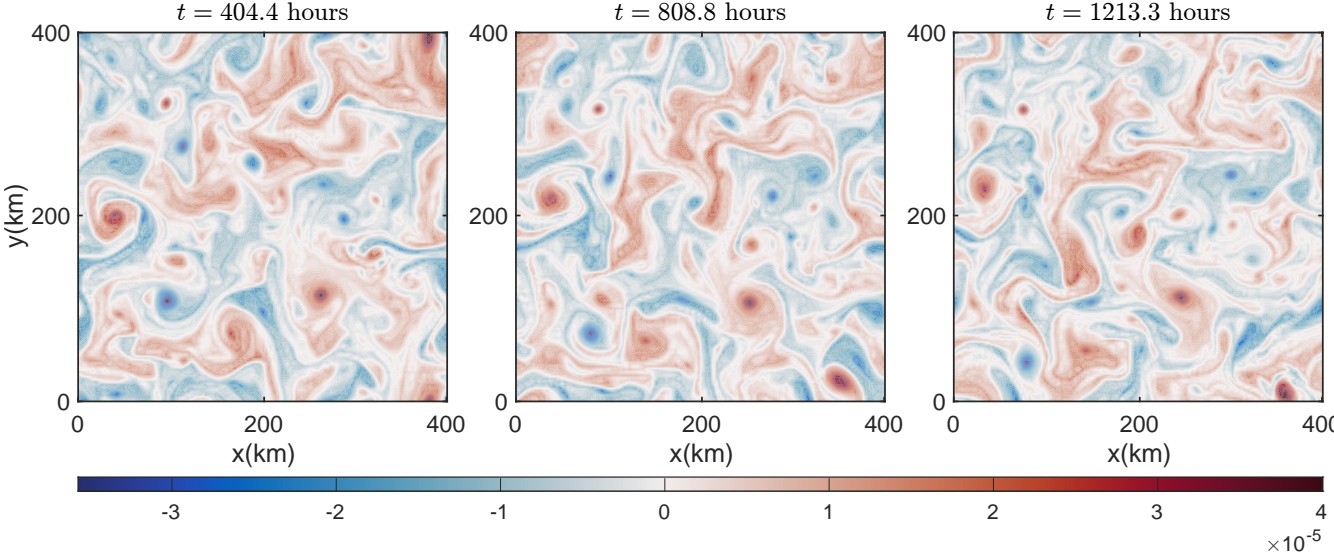

**Figure 3.** $PV_2$ field for the upper-layer ocean at different time instances.

### 4.3 Simulated floe trajectories

The trajectories presented in Figure 4 help understand the role of floe mass in modulating the sea ice motions. The floes have decreasing mass from top to bottom. The results here illustrate the relationship between floe size, mass, and external forces, such as wind stress and ocean currents.

The largest floe is shown in the top panel. The floe has a higher inertia and exhibits a smoother and more streamlined trajectory. The results in this panel suggest that, under similar forcing conditions, larger and heavier floes resist rapid changes in motion and respond more gradually to external perturbations due to their momentum. These floes dominate ice dynamics over longer time scales. They illustrate slower and more predictable moving patterns.

In contrast, the medium-sized floe in the middle panel shows a more irregular motion, with noticeable deviations from a
linear path. This is because its reduced mass allows for more immediate responses to short-term variations in external forcing, such as fluctuations in atmospheric wind or ocean currents. A balance between inertia and responsiveness appears in the motion of this floe. It highlights that mid-sized floes can exhibit complex dynamics even under relatively steady forcing conditions.

The bottom panel shows the trajectory of a small-sized floe. Due to the low mass and inertia of the floe, it has the strongest irregular motion. Therefore, the motion of small-sized floe is highly susceptible to external forces. Such floes contribute to the
more chaotic aspects of sea ice movement, especially in regions with strong atmospheric or oceanic variability.

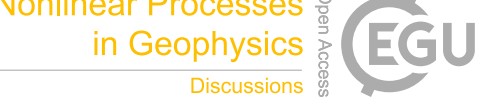



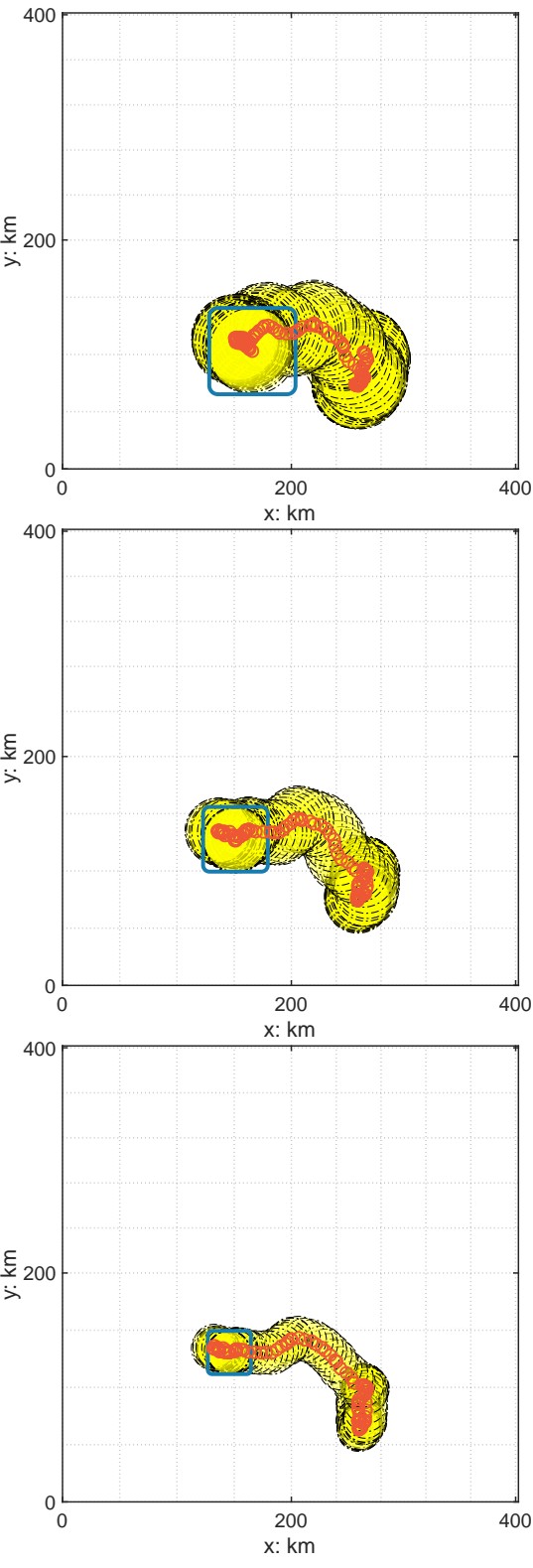

**Figure 4.** Trajectories of the largest floe in three different regimes (I, II, and III, from top to bottom), all starting from the same initial location, marked by a blue rectangle.





## 4.4 Simulated precipitation in the sea-ice-atmosphere coupled model

From a modeling perspective, accurately capturing the differences in evaporation rates between ice-covered and open ocean areas is essential for simulating the coupled atmosphere-sea ice system. The model must account for the spatial variability introduced by ice floes, including their size, distribution, and movement, as these factors significantly influence local evaporation
rates.

We calculate the spatially averaged total water content for each individual ice floe as follows:

$$[q_t]_l(t) = \frac{1}{|\Omega_l|} \int_{\Omega_l} q_t(x, y, t) d\mathbf{x}. \tag{55}$$

Here, $[q_t]_l(t)$ represents the average total water content over the area of the $l$-th ice floe at time $t$. The integral is taken over the domain $\Omega_l$, which denotes the spatial extent of the $l$-th floe, with $|\Omega_l|$ representing the area of this ice floe.





**Figure 5.** Floe distribution at different times for three different regimes top to bottom: Regime I, Regime II and Regime III, with different quantities: first column: floe distribution at $t = 808.8$ hours; second to third columns: total water at $t = 808.8$ and $1213.3$ hours; fourth column: averaged total water at $t = 808.8$ hours.

By integrating over the floe's area, we obtain a representative measure of the total water content, which can be used to analyze the floe's contribution to the overall hydrological balance in the sea ice-atmosphere system.

The first three columns of Figure 5 display the ice floes and total water content across different regimes. In the last column of Figure 5, the time-averaged spatially averaged total water content, $\overline{[q_t]_l(t)}$, is plotted as a function of different floe radii. The results demonstrate that $\overline{[q_t]_l(t)}$ decreases as the floe radius increases. This trend suggests that larger ice floes tend to have

lower average water content due to reduced moisture exchange.





# 5 Numerical Results of DA

## 5.1 Setup

The floe locations and upper atmospheric data are the only observational inputs used in the DA process for the coupled system.

The forecast time step, $\Delta t$, is set to 58.2 seconds, allowing the model to resolve finer temporal details. Observations are
available every 1500 numerical integration time steps, which corresponds to approximately $\Delta t^{obs}$ =24.2 hours. This frequency aligns with the acquisition of satellite images, which occurs roughly every 24 hours. The observational uncertainty for the ice floe locations, when there is no cloud cover, is set to 1 km. In contrast, the noise in the streamfunction is quantified as 20% of the standard deviation of the streamfunction at a given grid point, denoted as $\text{std}(\psi_{\mathbf{x}})$. The total duration of the DA spans 1601.5 hours, or approximately 66.73 days. While the truth is computed from a high-resolution model with $128 \times 128$ grids,
the reduced-order forecast model in spectral space contains $16 \times 16$ modes to enhance computational efficiency. The ensemble size in the Local Ensemble Transform Kalman Filter (LETKF) is 300. The localization radius is set to 200 km, which limits the influence of observations to a localized region, thereby reducing spurious correlations in the DA. The parameters used in the DA are summarized in Table 5.

| Parameter | Value |
|---|---|
| Observational time step $\Delta t^{obs}$ | 24.2 hours |
| Forecast time step $\Delta t$ | 58.2 s |
| DA time $T$ | 1601.5 hours $\approx 66.73$ days |
| Coarse grid points | $16 \times 16$ |
| Ensemble size | 300 |
| Localization radius | 200 km |
| Observational noise in floe trajecotries | Equation (50) |
| Observational noise in streamfunction | 20% of $std(\psi_2(\mathbf{x}))$ |

**Table 5.** Parameters in the DA

In the numerical tests, we consider two different settings for the observations of ice floes: (1) plentiful observations and (2)
sparse observations. In case (1), most of the floes ($\geq 70\%$)in the domain are observed during the observation period, whereas in case (2), only a few floes (about $30\%$) are observed during the same period.

## 5.2 Recovered atmosphere and ocean fields

### 5.2.1 Recovered atmosphere fields

Figure 6 presents the performance of DA in recovering the streamfunctions of the atmosphere flow fields, which are crucial in
describing the large-scale circulation of the atmosphere, at two different layers: the upper-layer streamfunction, $\psi_2$ (top panel), and the near-surface layer streamfunction, $\psi_1$ (bottom panel). The comparison is at $t = 970.61$ hours.



Despite the noisy observations, the posterior mean of the upper-layer streamfunction, $\psi_2$, closely matches the truth. The recovered near-surface streamfunction, $\psi_1$, shows slight deviations from the truth because $\psi_1$ is not directly observed. Its inference relies on a combination of observed floe trajectories and $\psi_2$, which introduces some uncertainty into the recovery of $\psi_1$. Nevertheless, despite the slight increase in uncertainty, the spatiotemporal patterns and amplitudes are largely recovered in the DA solution, demonstrating the effectiveness of DA in estimating unobserved states.

**Figure 6.** Recovered atmosphere streamfunction at the upper-layer, $\psi_2$ (top panel), and near-surface layer, $\psi_1$ (bottom panel) at $t = 970.61$ hours: comparison between the truth, and the posterior mean, and upper-layer observations.





### 5.2.2 Recovered upper-layer ocean field

Figure 7 shows the performance of DA in recovering the upper-layer ocean streamfunction. The figure compares the true ocean streamfunction (top panel) with the posterior mean (bottom panel) at several time instances.

The DA effectively restores the magnitude of the ocean fields and captures some of the large-scale patterns. The absence of small-scale features in the DA results is expected, as floe movements are primarily driven by atmospheric forces, with ocean drag playing a relatively minor role. Consequently, the ocean field has limited observability, resulting in less accurate state estimation.

**Figure 7.** Recovered upper-layer ocean streamfunction: comparison between the truth (top panel) and the posterior mean (bottom panel) at different time instances.





## 5.3 Recovered floe trajectories

Figure 8 compares the true and estimated trajectories of three ice floes under sparse observational conditions in Regime II over a period of 66.7 days. Similar results were observed in the other two regimes, so they are omitted here. In each panel, the posterior mean trajectory is shown in green, the observations in red, and the true trajectory in blue. The observational uncertainty, introduced via (50), shows that when cloud cover, $\widetilde{q}_t$, remains below a certain threshold, the uncertainty stays small (around 500 m), allowing for accurate floe location recovery. However, when cloud cover exceeds this threshold, the

observational uncertainty increases significantly, which can lead to substantial inaccuracies in the data assimilation of floe trajectories.

Panel (a) in Figure 8 shows the trajectory of an ice floe with a radius of 22.73 km, which is relatively large for this regime (see Figure 5). The results demonstrate a reasonably close match between the posterior mean and the true trajectory, despite sparse observational data. This is expected, as the value of $\overline{[q_t]_l(t)}$ is smaller for larger floes (as shown in the averaged total

water content in the second row of Figure 5), leading to lower observational uncertainty for floes with larger radii.

Panel (b) shows the trajectory of an ice floe with a radius of 22.03 km, slightly smaller than the one in Panel (a), which may lead to greater observational uncertainty at certain times. During these instances, the red observational points deviate significantly from the true trajectory. Nonetheless, the data assimilation (DA) still successfully recovers the floe trajectory, despite some highly inaccurate observations.

Finally, Panel (c) illustrates the trajectory of an ice floe with a radius of 12.23 km, significantly smaller than those in Panels (a) and (b). In this case, $\widetilde{q}_t$ remains relatively high throughout, indicating larger observational uncertainties. Consequently, the red observation points deviate significantly from the true locations at most observational time instances. Although the data assimilation (DA) recovered trajectory is less accurate compared to the previous two cases, it still captures the general trajectory pattern relative to the true path.



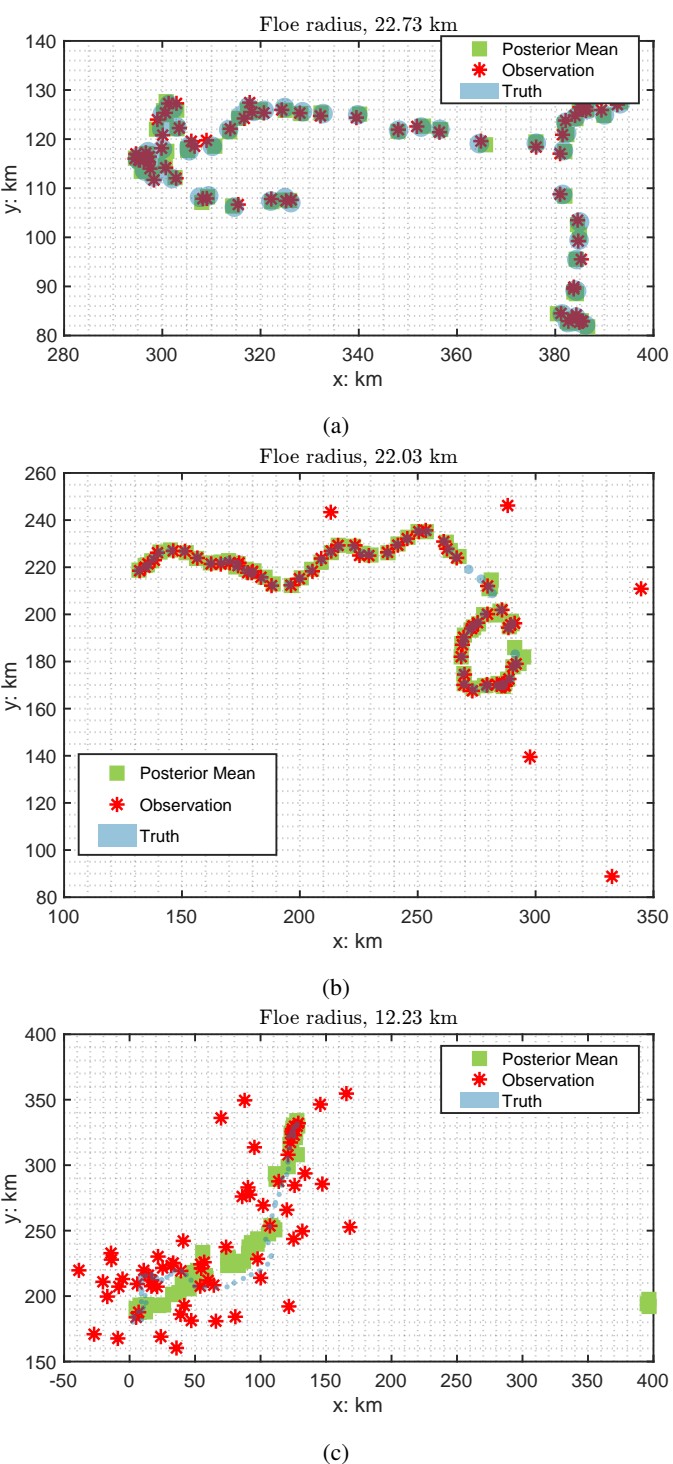

**Figure 8.** DA results for the ice floe trajectories: Panels (a)–(c) depict the floes' trajectories over time with the posterior mean (green), observations (red), and the true trajectory (blue).





## 5.4 Skill scores

Finally, we present the normalized root-mean-square error (RMSE) of the posterior mean estimate as a skill score for data assimilation (DA) across different regimes. The state variables in this DA study consist of: (1) the locations of the ice floes, and (2) the streamfunctions of the upper and near-surface atmosphere, as well as the ocean. Since the ice floe positions are Lagrangian quantities, while the streamfunctions are represented on Eulerian grids, the normalized RMSE skill scores are defined differently for the floe trajectories $\mathbf{x}$ and the streamfunction $\psi$:

$$
\begin{aligned}
RMSE_{\mathbf{x}} &= \frac{\sqrt{\frac{1}{M}\sum_{k=1}^{M}|\mathbf{x}^{\text{truth}}(t_k) - \mathbf{x}^{\text{posterior}}(t_k)|^2}}{\sqrt{\frac{1}{M}\sum_{k=1}^{M}|\mathbf{x}^{\text{truth}}(t_k)(t_k)|^2}} \\
RMSE_{\psi} &= \frac{\sqrt{\frac{1}{NM}\sum_{j=1}^{N}\sum_{k=1}^{M}|\psi^{\text{truth}}(\mathbf{x}_j,t_k) - \psi^{\text{posterior}}(\mathbf{x}_j,t_k)|^2}}{\sqrt{\frac{1}{NM}\sum_{j=1}^{N}\sum_{k=1}^{M}|\psi^{\text{truth}}(t_k)(\mathbf{x}_j,t_k)|^2}}
\end{aligned}
\tag{56}
$$

where $N$ is the number of grid points and $M$ is the number of observation, $(\cdot)^{\text{truth}}(t_k)$ is the true position of the floes at time $t_k$, and $(\cdot)^{\text{posterior}}(t_k)$ represents the model's posterior mean estimate of the floe position at time $t_k$. When the normalized RMSE exceeds 1, it indicates a loss of skill in the estimation, as the error in the posterior mean reaches the level of the equilibrium standard deviation.

The DA skill scores across the three regimes are shown in Tables 6–8. For the fully observed atmospheric state variable, $\psi_2^a$, and the unobserved variable, $\psi_1^a$, the DA demonstrates strong performance. This is expected, as $\psi_1^a$ and $\psi_2^a$ are closely linked by barotropic and baroclinic dynamics, allowing for relatively accurate recovery of $\psi_1^a$ even though it is not directly observed. In contrast, the estimation of the ocean streamfunction, $\psi^o$, is less accurate. This is because in the coupled model, the floe movements are primarily driven by atmospheric forces, while ocean drag has a relatively weak influence. As a result, the ocean field has limited observability, leading to less accurate state estimation. On the other hand, the estimation of floe trajectories is generally accurate, with accuracy slightly decreasing from high-concentration (Regime I) to low-concentration regimes (Regime III). This is due to the lower concentration regime experiencing more cloud cover (Figure 5), which amplifies observational uncertainties.





| Observation level | State Variable | Normalized RMSE |
|---|---|---|
| Plentiful Observation | $\psi_2^a$ | $9.1702e-02$ |
| | $\psi_1^a$ | $4.8378e-01$ |
| | $\psi^o$ | $1.8559e+00$ |
| | $\mathbf{x}_l$ | $3.2733e-02$ |
| Sparse Observation | $\psi_2^a$ | $7.9300e-02$ |
| | $\psi_1^a$ | $6.1109e-01$ |
| | $\psi^o$ | $1.8051e+00$ |
| | $\mathbf{x}_l$ | $1.6376e-01$ |

**Table 6.** Normalized RMSE of the DA results in Regimes I.

| Observation level | State Variable | Normalized RMSE |
|---|---|---|
| Plentiful Observation | $\psi_2^a$ | $8.3388e-2$ |
| | $\psi_1^a$ | $5.3918e-1$ |
| | $\psi^o$ | $1.6619e0$ |
| | $\mathbf{x}_l$ | $4.5736e-2$ |
| Sparse Observation | $\psi_2^a$ | $8.0821e-2$ |
| | $\psi_1^a$ | $5.4577e-1$ |
| | $\psi^o$ | $1.7079e0$ |
| | $\mathbf{x}_l$ | $2.4368e-1$ |

**Table 7.** Normalized RMSE of the DA results in Regimes II.

| Observation level | State Variable | Normalized RMSE |
|---|---|---|
| Plentiful Observation | $\psi_2^a$ | $8.0383e-02$ |
| | $\psi_1^a$ | $5.2187e-01$ |
| | $\psi^o$ | $1.8353e+00$ |
| | $\mathbf{x}_l$ | $8.1702e-02$ |
| Sparse Observation | $\psi_2^a$ | $7.5127e-02$ |
| | $\psi_1^a$ | $5.0754e-01$ |
| | $\psi^o$ | $1.7287e+00$ |
| | $\mathbf{x}_l$ | $5.4736e-01$ |

**Table 8.** Normalized RMSE of the DA results in Regimes III.





# 6 Conclusion

In this paper, we developed an idealized coupled atmosphere-ocean-ice model to investigate the effects of clouds on sea ice dynamics in the MIZ. Our model integrates the DEM to simulate the movement and interaction of individual ice floes, combined with a two-layer QG ocean model and a two-layer PQG atmospheric model that includes saturated precipitation processes. This framework facilitates studying the interactions between atmospheric, ocean, and sea ice floes. It specifically focuses on cloud-induced radiative and precipitation effects on sea ice evolution. The paper addresses both forward (model simulation) and inverse (DA) problems. For the former, we study the interactions between different model components; for the latter, we focus on recovering unobserved floe trajectories obscured by cloud cover and inferring ocean and atmospheric fields using limited observations.

The results from this study show the significant influence of clouds on sea ice dynamics. They also help understand the non-trivial interactions within the coupled atmosphere-ocean-ice system. Integrating idealized modeling with advanced DA techniques forms a powerful tool for enhancing the prediction of Arctic climate processes, particularly in the MIZ, where small-scale interactions are critical.

There are several future research works. One is about further refining the model to incorporate more sophisticated thermodynamic processes, and the other is to test its performance under varying climate scenarios. Moreover, the continued development of DA schemes that can handle more comprehensive observational datasets Curry et al. (1996) will be essential for advancing our understanding of sea ice dynamics and for improving the accuracy of Earth system models.

*Author contributions.* C.M., N.C., and S.S. contributed to the design and implementation of the research, to the analysis of the results and to the writing of the manuscript.

*Competing interests.* The authors declare that they have no conflict of interest.

*Acknowledgements.* The research of S.N.S and N.C. is partially funded by the Office of Naval Research (ONR) Multidisciplinary University Research Initiative (MURI) award N00014-19-1-2421. C.M. was supported as a postdoc research associate under this grant.





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

## Appendix A:  Contact forces in the coupling model

### A0.1  Normal contact force

In the mathematical framework described in equations (36) and (5), the normal contact force between the $l$-th and $j$-th ice floes is governed by Hooke's law of linear elasticity. Specifically, the force $\mathbf{f_n}^{lj}$ is expressed as:

$$\mathbf{f_n}^{lj} = c^{lj} E^{lj} \delta n^{lj} \mathbf{n}^{lj}, \tag{A1}$$

where $\mathbf{n}^{lj}$ denotes the normal vector originating from the center of floe $l$ and pointing towards the center of floe $j$. The function

$\delta_n^{lj}$, representing a Delta function, ensures that the contact force between the $j$-th and the $l$-th floes is nonzero only when they are in contact:

$$\delta_n^{lj} = \begin{cases} 1 & \text{if } |d^{lj}| - (r^l + r^j) < 0, \\ 0 & \text{otherwise.} \end{cases} \tag{A2}$$

Here, $r^l$ and $r^j$ are the radii of the respective floes, and $d^{lj}$ is the distance between their centers. The chord length $c^{lj}$, oriented transversely across the cross-sectional area, is calculated as:

$$c^{lj} = \frac{1}{d^{lj}} \sqrt{4(d^{lj})^2 (r^{max})^2 - ((d^{lj})^2 - (r^{min})^2 + (r^{max})^2)^2} \tag{A3}$$

where $E^{lj}$ represents the Young's modulus of elasticity. The parameters $c^{lj}$ and $d^{lj}$, crucial for understanding the mechanical interactions, are illustrated in Figure A1.

### A0.2  Tangent contact force

The tangential force at the contact interface between two ice floes is proportional to their relative tangential velocities. The

direction of this force, denoted by $\mathbf{t}^{lj}$, is perpendicular to the normal direction. This force, $\mathbf{f_t}^{lj}$, is defined by the following expression:

$$\mathbf{f_t}^{lj} = c^{lj} G^{lj} \Delta v_t^{lj} \mathbf{t}^{lj}, \tag{A4}$$

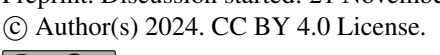


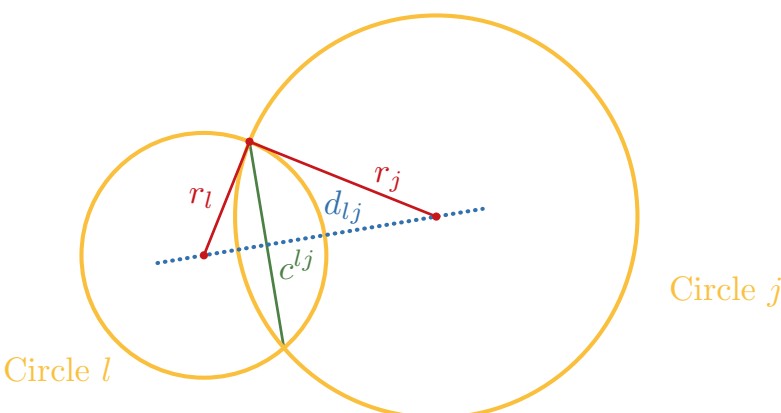

**Figure A1.** Illustration of geometrical quantities for computing the tangential force and the normal force.

where $c^{lj}$ refers to the chord length previously defined in (A3), $G^{lj}$ is the shear modulus, and $\Delta v_t^{lj}$ is the difference in velocity along the tangential direction between the $l$-th and $j$-th ice floes.