# Peer review of "Simulation and Data Assimilation in an Idealized Coupled Atmosphere-Ocean-Sea Ice Floe Model with Cloud Effects"

_Nonlinear Processes in Geophysics, 2024_

## Author Comment (AC1)

**Response to Referee #1**

Changhong Mou, Samuel N Stechmann, Nan Chen\*

[May 29, 2025]

**1 General Comments**

The manuscript is well written, model equations are documented well. I found Section 3 a bit hard to follow since the majority of it describes surrogate models not DA. But it is fine since those are used only in the ensemble DA experiments. However, the DA system description is left limited; this may make the manuscript hard to follow for non-expert readers on DA while looking at the experiments in Section 5.

Furthermore, the validation of the experiments can be done in a more quantitative way. The skill of the posterior mean is documented in Tables 6-8 but they can be extended to prior mean to see how much the DA improves with respect to the forecast. Another way of doing it is using the non-assimilative experiments in Section 4 as a baseline to see how the sparse and plenty observation cases improve when observations are incorporated. Another interesting diagnostic can be to look at the spread of the prior and posterior ensemble to see the uncertainty before and after the assimilation.

Finally, conclusions can be extended considering my specific comments below. Overall, I would expect a more quantitative assessment of the results. I would be happy to see a revised version.

**Response:** We sincerely thank the reviewer for the detailed comments and very helpful suggestions. We have carefully addressed each comment and made the changes accordingly throughout the revised manuscript. The details are as follows.

**2 Specific Comments**

Introduction: The data assimilation literature related to the work should be extended to give the reader a view of which studies have been done in line with the investigation presented in Section 5.

We thank the reviewer's comment and we agree that we need to add the literature review of data assimilation in the related area. In the revised manuscript, we have added the following additional literature overview in the section of introduction:

"Recent advances in sea-ice data assimilation encompass a broad spectrum of approaches. For example, [Lisæter et al., 2003, Massonnet et al., 2014] assimilated passivemicrowave concentration and altimetry-derived thickness into coupled ice-ocean models with an ensemble Kalman filter, substantially reducing drift and thickness errors. [Riedel and Anderson, 2024] accounted for the bounded, non-Gaussian statistics of seaice variables within the observation operator, which refines the posterior analyses of both ice and snow states. At the fully coupled level, [Penny et al., 2019] introduced a strongly coupled data-assimilation (SCDA) framework that puts sea-surface and ice increments directly into the atmospheric analysis, further improving the short-term forecasts in the marginal-ice zone. With the traditional Eulerian approaches, [Chen et al., 2022b, Deng et al., 2025] developed an efficient Lagrangian scheme that reconstructs mesoscale currents and vorticity from a limited set of tracked floes, even if only partial trajectories are observed due to clouds. Nevertheless, current data-assimilation frameworks for fully coupled atmosphere-ocean-ice models still lack a consistent treatment of cloud and precipitation effects. "

Section 2: Sections 2.2.4 and 2.2.4 can be subsections of 2.2.2. Similarly, 2.2.6 and 2.27 can be subsections of 2.2.5.

**Response:** We thank the reviewer for the helpful suggestion. We have combined Subsections 2.2.2–2.2.4 into one subsection titled "Oceanic forcing", and Subsections 2.2.5–2.2.7 into "Atmospheric forcing".

Section 3: how costly is the coupled system? Can you be a bit precise in terms of cpu time? Is the code parallelized? What is the gain with the reduced-order models in this case?

**Response:** We thank the reviewer for the question. We have revised Section 3 to clarify the computational cost of the coupled model and the role of the reduced-order model (ROM).

The full coupled atmosphere–ice–ocean model is implemented in a sequential MATLAB code and is *not* parallelized. A single forward run over one simulated day (i.e., 86,400 seconds) requires approximately 3.70 CPU-hours, corresponding to a wall-clock time of 0.82 hours on an Apple M1 Max CPU with 32 GB of RAM. This computational cost makes ensemble-based data assimilation prohibitively expensive for even with a small ensemble size. While a parallelized implementation could mitigate this computational burden, it would require substantial effort in restructuring the code and managing inter-process communication, which is beyond the scope of this study.

By contrast, the proposed reduced-order models (ROMs) for the coupled system are several orders of magnitude faster and enable efficient ensemble forecasting. In addition, we aim to demonstrate the effectiveness of data assimilation using standard ROMs that preserve the essential system dynamics under observation scenarios where ice floes are irregularly obscured by cloud cover.

In line 265 of the revised manuscript, we have added the following:

For instance, the full coupled atmosphere-ice-ocean model is implemented as a sequential MATLAB code without parallelization. A single forward simulation over one model day (i.e., 86,400 seconds) requires approximately 3.70 CPU-hours, corresponding to a wall-clock time of 0.82 hours on an Apple M1 Max processor with 32 GB of RAM. It is therefore not feasible to use this full order model directly in data assimilation settings due to its high computational cost. Section 3.2 it is not clear to me how the total water content is linked to the ice floe area?

**Response:** We thank the reviewer for the careful reading. The link between ice-floe area and the total water content  $q_t$  operates in two ways. *First*,  $q_t$  is a prognostic variable in the atmospheric model that does not directly relate to the sea-ice area. However, the ice floe area has a different evaporation rate, which contributes to different water transport from the ocean surface, and this can contribute to the different distribution of the total water over the ice floe and the ocean. Some numerical results are discussed in Section 4.4. *Second*, the mean total water content  $[q_t]$  over the *l*th ice floe is used as a quantitative indicator for cloud cover: larger  $[q_t]$  indicates thicker or more extensive clouds, which obscure ice-floe edges and therefore is represented by larger observation uncertainties in floe position. Section 3.2 details how this uncertainty is incorporated.

In the end of Section 3.2, we have added the following to better illustrate the link:

In summary, the total water content  $q_t$  interacts with ice floes in two different ways. First, the spatial coverage of an ice floe over the ocean affects the local evaporation, which in turn modifies the distribution of  $q_t$  over ice and ocean. Second, the mean total water content  $[q_t]$  above each floe is used to parameterize cloud-related observation uncertainty in floe localization, with higher  $[q_t]$  indicating thicker clouds and thus higher uncertainty.

Using "q" in the equations for both the PV and water content is confusing from time to time. Section 3.3.2 describes directly the data assimilation scheme (LETKF) not the localizations. This can be the introduction to Section 3.3. You don't detail the scheme, no KF equations for example, but you allocate a dedicated section to localization. What is the reason that you prefer to discuss localization explicitly for your application? How does your analysis benefit from it?

**Response:** We thank the reviewer for pointing out the potential confusion arising from using the symbol q. We have chosen to retain this notation in order to be consistent with prior work, particularly the formulation in [Qi and Majda, 2016, Chen et al., 2021, Chen et al., 2022a], which uses q for both quantities for potential vorticity. To clarify the meaning in our manuscript, we have added one sentence in the line 160 of section 2.3 where this notation first appears to remind the reader of the specific interpretation of q based on the context:

Here,  $q_{(.)}$  denotes the potential vorticity at the (.) layer, following the notation used in [Qi and Majda, 2016, Chen et al., 2021, Chen et al., 2022a]. Its precise definition may vary slightly depending on the component (e.g., atmosphere or ocean) and is clarified in the corresponding context.

For the second point, we thank the reviewer for this thoughtful question. While the ensemble Kalman filter (EnKF) and its variants such as LETKF are well-established in the data assimilation (DA) literature such as [Evensen, 2009, Asch et al., 2016], we chose to put a dedicated section to localization due to its central importance in our application. Specifically, our coupled model features both Eulerian state variables (e.g., atmospheric and oceanic streamfunctions on a grid) and Lagrangian variables (i.e., individual ice floe trajectories), which evolve on different spatial frameworks and display different spreads of uncertainties. Therefore, localization plays a critical role in assimilating sparse observations in systems with mixed Eulerian–Lagrangian information.

By explicitly detailing our approach to localization, we aim to give the audience a clear presentation about how these different kinds of state variables are combined in the localization in data assimilation. Thus, while the LETKF algorithm is standard and well-known, the localization for different state variables in this work is both non-trivial and essential to the effectiveness of the assimilation procedure. For this reason, we believe it merits a focused discussion.

To make it clear, we have revised line 432 in the draft:

LETKF applies filtering exclusively in physical space for both the trajectories of ice floes which are in the Lagrangian frame of reference and the streamfunctions of the atmosphere and ocean that are in the Eulerian frame of reference.

Section 4.1 Is the time step equal for all the components? If so, what are the implications for resolved scales for the supposedly fast evolving atmosphere and relatively slow ocean and sea ice? By the way, is there a specific reason that you choose the timestep as a decimal number in seconds? It is 58.2 seconds in line 399 while 1.2941 hours in Table 1. Not clear to me why they are different.

**Response:** We thank the reviewer for pointing this out. In our simulation, three distinct time steps are employed:

- Atmospheric model:  $\Delta t = 58.2 \,\mathrm{s}$
- Ocean model:  $\Delta t = 1.2941$  hours
- Ice floe model:  $\Delta t = 1.2941$  hours

These dimensional time steps are derived by rescaling from their non-dimensional counterparts based on characteristic scales appropriate for the Arctic regime used in our study. For example, the atmospheric model's dimensional time step of 58.2s corresponds to the non-dimensional value  $\Delta t = 5 \times 10^{-4}$ , given the Arctic characteristic time scale T used in the QG model.

In addition, we have intentionally chosen different time steps for the three models to significantly reduce the computational cost. The primary computational burden in the coupled model arises from the ice floe component, which must resolve collisions as well as contact forces and torques among floes at every time step. If the ice floe model were computed using the atmospheric model's smaller time step, the computational expense would increase by approximately 80 times. Therefore, employing different time steps for each model is an effective strategy to balance computational efficiency and model accuracy.

To make this clearer, we have updated the time scale and time step information in Tables 2, 3, and 4, as well as added the following to the Section 4.1 of the manuscript:

It is worth noting that the dimensional time steps for the atmospheric and ocean models are obtained by rescaling their respective non-dimensional counterparts using their different characteristic scales appropriate to each component under the Arctic regime; for example, the atmospheric model's time step of 58.2s corresponds to a non-dimensional value of  $\Delta t = 5 \times 10^{-4}$ , based on the characteristic time scale T employed in the PQG model. Section 4.3 The trajectory of the large floes seems to me more unpredictable including returns and changing directions. Is there a way to quantify this behavior?

**Response:** We thank the reviewer for highlighting this insightful observation. Indeed, large ice floes exhibit more complex trajectories characterized by frequent directional changes and occasional reversals. A quantitative assessment of this intricate behavior could involve analyzing metrics such as path curvature, directional persistence, or employing methods from stochastic trajectory analysis. However, a detailed quantification of these aspects lies beyond the scope of the current study and will be investigated systematically in our future work. In the revised manuscript, we have added one sentence:

In addition, it would also be worthwhile to explore how the statistical properties of ice floe trajectories vary with floe size—while some aspects of this behavior, particularly for smaller floes, are illustrated through case studies in this work, a more systematic quantification is left for future investigation.

Section 5.1 Returning back to my comment on Section 3, did you try running an experiment with smaller ensemble size (instead of 300) using the full models (instead of surrogates)?

**Response.** We appreciate the reviewer's interest in a baseline test that would employ the full coupled model rather than the surrogate emulator. Unfortunately, even with a much smaller ensemble, such an experiment is prohibitively expensive:

- A single 66-day forecast with the full atmosphere–ocean–ice model costs approximately 244 CPU-hours. Running an ensemble of 50 members—the minimum size that still yields a well-conditioned sample covariance for this state dimension—would therefore require about 12200 CPU-hours per analysis cycle.
- The full coupled model is discretized on a much finer spatial grid than the surrogate (ROM), yielding a state vector whose dimension is orders of magnitude larger. Assimilating such high-dimensional state variables would take a substantial computational cost for each analysis cycle; even a minimum setup would be infeasible for a few data assimilation cycles in Section 5.1.

For these reasons we opted to use surrogate dynamics in combination with a larger ensemble (300 members) to preserve covariance accuracy at manageable cost. A systematic comparison between a ROM based ensemble and an FOM based ensemble in data assimilation is an important topic for future work once additional computational resources become available.

To make it clear about the computational cost for DA with the full order model, we have added the following statement at the beginning of Section 3 in the revised manuscript:

While it would be ideal to perform a baseline data assimilation experiment using the full coupled atmosphere-ocean-ice model, such a setup is computationally prohibitive. A single 66-day forecast with the full model requires approximately 244 CPU-hours, and an ensemble of 50 members, which is a relatively small ensemble size, would require over 12,000 CPU-hours per assimilation cycle. Furthermore, the full model operates on a much finer spatial grid than the reduced-order surrogate, resulting in a substantially higher-dimensional state variable and significantly more expensive computational costs.

What are the state variables, all model variables? Section 5.2 It would be useful to provide the prior mean and spread of both the analysis and forecast to assess the improvements via DA. The posterior mean compared to truth is fine but doesn't show how much the state and the trajectory improved.

**Response.** We thank the reviewer for the helpful suggestion.

The *state variables* in our coupled atmosphere–ocean–sea ice model include:

- the streamfunctions for the upper and lower atmosphere layers, denoted by  $\psi_1^a$  and  $\psi_2^a$ ,
- the streamfunction for the surface ocean,  $\psi_2^o$ , and
- the two-dimensional positions of each individual ice floe at time t, represented by  $(x_l(t), y_l(t))$  for floe index l.

In response to the second point, we have revised Section 5.2, along with Figures 6–8 and Tables 6–8, to include both the *prior* (forecast) and *posterior* (analysis) means for the atmospheric and oceanic state variables, as well as the ice floe trajectories.

**3 Technical corrections**

Format of the citations are not adequate and should be corrected all over the text. e.g. Cámara-Mor et al. (2010); Kwok (2018)  $\rightarrow$  (Cámara-Mor et al. 2010; Kwok 2018)

**Response:** We thank the reviewer for pointing this out. We have revised and corrected the citation format in the manuscript.

**References**

- [Asch et al., 2016] Asch, M., Bocquet, M., and Nodet, M. (2016). Data assimilation: methods, algorithms, and applications. SIAM.
- [Chen et al., 2021] Chen, N., Fu, S., and Manucharyan, G. (2021). Lagrangian data assimilation and parameter estimation of an idealized sea ice discrete element model. *Journal of Advances in Modeling Earth Systems*, 13(10):e2021MS002513.
- [Chen et al., 2022a] Chen, N., Fu, S., and Manucharyan, G. E. (2022a). An efficient and statistically accurate lagrangian data assimilation algorithm with applications to discrete element sea ice models. *Journal of Computational Physics*, 455:111000.
- [Chen et al., 2022b] Chen, N., Fu, S., and Manucharyan, G. E. (2022b). An efficient and statistically accurate lagrangian data assimilation algorithm with applications to discrete-element sea-ice models. *Journal of Computational Physics*, 456:111000.
- [Deng et al., 2025] Deng, Q., Chen, N., Stechmann, S. N., and Hu, J. (2025). Lemda: A lagrangianeulerian multiscale data assimilation framework. *Journal of Advances in Modeling Earth Systems*, 17(2):e2024MS004259.

- [Evensen, 2009] Evensen, G. (2009). Data assimilation: The ensemble Kalman filter. Springer Science & Business Media.
- [Lisæter et al., 2003] Lisæter, K. A., Rosanova, J., and Evensen, G. (2003). Assimilation of ice concentration in a coupled ice–ocean model using the ensemble kalman filter. Ocean Dynamics, 53:368–388.
- [Massonnet et al., 2014] Massonnet, F., Goosse, H., Fichefet, T., and Counillon, F. (2014). Calibration of sea-ice dynamic parameters in an ocean–sea-ice model using an ensemble kalman filter. Journal of Geophysical Research: Oceans, 119(7):4168–4184.
- [Penny et al., 2019] Penny, S. G., Hamill, T. M., Akella, S., et al. (2019). Strongly coupled data assimilation in multiscale media: Experiments using a quasi-geostrophic coupled model. *Journal* of Advances in Modeling Earth Systems, 11:1803–1829.
- [Qi and Majda, 2016] Qi, D. and Majda, A. J. (2016). Low-dimensional reduced-order models for statistical response and uncertainty quantification: Two-layer baroclinic turbulence. *Journal of* the Atmospheric Sciences, 73(12):4609–4639.
- [Riedel and Anderson, 2024] Riedel, C. and Anderson, J. (2024). Exploring non-gaussian sea-ice characteristics via observing system simulation experiments. *The Cryosphere*, 18:2875–2896.

---

## Author Comment (AC2)

**Response to Referee #2**

Changhong Mou, Samuel N Stechmann, Nan Chen*

[May 29, 2025]

**1 General Comments**

**Comment 1.**
The governing equations for sea ice floes do not appear to include the Coriolis force. Please clarify whether this was omitted intentionally and, if so, provide a justification.

**Response:** We thank the reviewer for this insightful remark. Indeed, the Coriolis force is not explicitly included in the governing equations for the sea ice floes presented here. This omission was intentional, as our current model primarily focuses on resolving short-term, localized ice floe dynamics dominated by collision and contact interactions. At these smaller spatial and temporal scales, the effects of the Coriolis force are considerably weaker compared to other forces, such as drag, collision, and contact forces (see, e.g., [Thorndike and Colony, 1982, Steele et al., 1997]; also Chapter 9, "Ice Dynamics," in [Thorndike, 1986]). Nonetheless, we acknowledge the potential significance of the Coriolis effect for larger-scale and longer-term ice floe motions, and incorporating this force explicitly will be considered in future studies to assess its impact comprehensively.

In the revised manuscript, we have added the following in the end of Section 2.2.1:

> *While the current coupling model does not include the Coriolis force on ice floe movement, it is still a valid assumption as this research focuses on short-term, localized ice floe dynamics where collision and drag forces are dominant [Thorndike and Colony, 1982, Steele et al., 1997, Thorndike, 1986].*

**Comment 2.**
Are the background velocities of ocean and atmosphere considered for the computation of drag forces and torques on the floes?

**Response:** We thank the reviewer for raising this question. Yes, the background zonal velocities $U^a$ and $U^o$ from the atmospheric and oceanic models are implicitly considered in calculating the drag forces and torques acting on the ice floes. Specifically, the QG models used to generate atmospheric and oceanic flows incorporate these imposed background zonal velocities $U^a$ and $U^o$, thereby implicitly influencing the resulting flow fields. Consequently, these background velocities directly affect the dynamic responses and trajectories of the ice floes in our simulations.

To clarify the role of background velocities, we have added Remark 2.1 to the manuscript:

**Remark 2.1** *The background zonal velocities $U^a$ and $U^o$ in the atmospheric and oceanic QG models are implicitly included in the computation of drag forces and torques on the ice floes. These background zonal velocities affect the flow fields, thereby influencing the ice floe dynamics and trajectories in the coupled system.*

**Comment 3.**
There isn't any coupling between atmosphere and ocean and the coupling between ice floes and ocean is only one-way. Can you justify this choice? This may have a significant impact on the accuracy of ocean state estimation, which is shown to perform relatively poorly in the data assimilation experiments.

**Response:** We thank the reviewer for this important comment. Indeed, in the current study, the coupling is limited: the atmosphere and ocean are not directly coupled, and the interaction between ocean and ice floes is one-way—from ocean to ice. This modeling choice was made to isolate and examine the impact of precipitation and its effects on the coupled system, particularly the response of sea ice floes to atmospheric and oceanic forcing under uncertain moisture conditions.

As an investigation of data assimilation in a coupled atmosphere–ice–ocean framework with precipitation, our goal is to build foundational understanding while keeping the system tractable. We acknowledge that omitting full two-way coupling—especially between ocean and ice—may affect the accuracy of ocean state estimation. However, our results still provide important insights into how observational uncertainty from moisture and precipitation propagates through the system.

Extending the current model to include two-way feedbacks between ice and ocean, as well as a fully coupled atmosphere–ocean interaction, is a natural and important direction for future work and will likely improve ocean state estimation in data assimilation settings. Therefore, we have added the following two modifications in the draft:

- We have added Remark 2.2:

    *In the presence of sea ice floes, the direct coupling between the atmosphere and ocean is relatively weak compared to other components of the system, such as the atmosphere–ice floe interaction. Moreover, the atmospheric and oceanic models have different time scales—specifically, the ocean evolves on a much slower time scale than the atmosphere—and the current study focuses on short-term simulations. As a simplification, we choose to neglect the atmosphere–ocean coupling, which remains a reasonable assumption within the scope of the present modeling framework.*

- In the Conclusion section, we have added the following statement:

    *As a future direction, we aim to extend the current framework by incorporating two-way coupling between the ocean and ice components, as well as a fully coupled atmosphere–ocean interaction, to better capture feedback mechanisms and improve the accuracy of ocean state estimation in data assimilation settings.*

**Comment 4.**
Line 229: it is assumed that precipitation causes an increase in ice floe thickness. This may be a reasonable assumption if precipitation is snow, but this is not guaranteed especially during summer and fall and in the MIZ [Boisvert et al. 2023]. Furthermore, solar insolation is set to 1361 W/m2 (Table 4), which is very unrealistic for polar winters.

**Response:** We thank the reviewer for this helpful comment. The assumption that precipitation increases ice floe thickness is based on the assumption of our model, where precipitation is interpreted as snowfall. We acknowledge that this assumption is most valid during colder months and in regions where snow dominates, and may not hold during summer or fall in the Marginal Ice Zone (MIZ), as highlighted in [Boisvert et al., 2023]. As a first step toward investigating the influence of precipitation on ice floe height, we adopt this assumption; in addition, the ice floe thickness changes does not significantly impact the overall dynamics of the coupled system. In the revised manuscript, we have added the following statement in the end of Section 2.5.1:

> *It is noted that the assumption that precipitation increases ice floe thickness is based on the idealization that all precipitation is interpreted as snowfall. This assumption is most valid during colder months and in regions where snow predominates, and may not hold during summer or fall in the Marginal Ice Zone (MIZ), as highlighted in [Boisvert et al., 2023]. As a first step toward investigating the influence of precipitation on ice floes, we adopt this assumption to explore its impact within an idealized modeling framework.*

Regarding the solar insolation value of 1361 W/m$^2$, we note that the chosen value corresponds to the solar constant and is used as a simplified representation of solar radiation to mimic an idealized melting process for ice floes. While this value is not realistic for polar winters, the simplification is reasonable given that our current study focuses on a short simulation period of approximately two months. It enables us to isolate and investigate key coupled dynamics in a controlled setting. In the manuscript, we have revised the following sentence in Section 2.5.2:

> *Here, $E_s$ represents the solar insolation, known as the solar constant. While this value may differ in polar winter conditions, it is treated as constant in this study as a simplification appropriate for the short simulation period.*

**Comment 5.**
Line 464: Clearly state that the "truth" is obtained from the full coupled model and describe the numerical scheme used to solve it.

**Response:** We thank the reviewer for this helpful suggestion. In the revised manuscript, we have explicitly stated in Line 464 that the "truth" trajectory is obtained by running the full coupled atmosphere–ice–ocean model using high-resolution numerical integration.

The numerical schemes employed for each component of the model are as follows: the atmospheric and oceanic equations are spatially discretized using a spectral method, and temporally integrated using an adaptive third-order Runge–Kutta scheme, following the approach in [Qi and Majda, 2016, Edwards et al., 2020a, Edwards et al., 2020b]. The sea ice component is simulated using the discrete element method (DEM), with time integration performed via a forward Euler scheme.

In the beginning of Section 4, we have added the following in the revised manuscript:

*The coupled atmosphere–ocean-ice model employs different numerical schemes for each component. The atmospheric and oceanic equations are discretized in space using a spectral method and integrated in time with an adaptive third-order Runge–Kutta scheme, following [Qi and Majda, 2016, Edwards et al., 2020a, Edwards et al., 2020b]. The sea ice component is simulated using the discrete element method (DEM), with time integration using a forward Euler scheme to resolve floe dynamics dominated by contact and drag forces.*

**Comment 6.**
Line 470: The definition of plentiful and sparse observations is not clear. Have you applied Equation (54) with different values of the threshold?

**Response:** We thank the reviewer for pointing out this. Equation (54) determines whether the position of the $l$-th floe is observed at an observation time $t_k$:

$$\begin{cases} q_{t,m}\left(\mathbf{x}_l^{\mathrm{obs}}(t_k)\right) < q_\epsilon & \implies \text{floe is } observed, \\ q_{t,m}\left(\mathbf{x}_l^{\mathrm{obs}}(t_k)\right) \geq q_\epsilon & \implies \text{floe is } not \text{ observed.} \end{cases} \tag{54}$$

Here $q_{t,m}$ is the total water that is from the atmospheric model at $m$th floe location at time $t$ and $q_\epsilon$ is a tunable threshold. By increasing $q_\epsilon$ we expect more floes are observed and thus have plentiful observations, whereas a lower $q_\epsilon$ admits less observed ice floes and yields sparse observations. Based on this principle, we choose two different regimes of $q_\epsilon$ which yield two different percentage of floes that are observed:

- **Plentiful observations**: at least 70% of the floes satisfy "observed condition" in Eq. (54) and therefore are observed;

- **Sparse observations**: only about 30% of the floes satisfy "observed condition" in Eq. (54) and thus are observed.

These explicit percentages—and the corresponding values of $q_\epsilon$—are now stated directly below Table 5 in the revised manuscript, clarifying how the terms "plentiful" and "sparse" are defined.

**Comment 7.**
Figure 6-7-8: It would be helpful to include the background mean for visual comparison with the analysis.

**Response:**
We thank the reviewer for this helpful suggestion. To make a more complete visual comparison, we have updated Figures 6–8 to include the background (prior) mean with the analysis (posterior) and truth. This update allows readers to better assess the improvement obtained from data assimilation by comparing the prior and posterior states relative to the true state.

**Comment 8.**
Line 485: The manuscript states that the ocean field has limited observability. Could you elaborate on this and possibly provide a more quantitative analysis?

**Response:** We agree that a systematic, quantitative assessment of ocean–state observability would be valuable; however, such an analysis would require dedicated sensitivity experiments (e.g. varying mooring density, tracks, and electromagnetic under-ice soundings) that lies beyond the scope of the present study, whose focus is the feasibility of assimilating floe trajectories under cloud "contamination". A rigorous observability study will be the subject of future work.

To avoid any ambiguity in the current manuscript, we have slightly rephrased the relevant sentence (Line 605) to read

> *"Because existing in-situ and remote sensors sample only limited portions of the stratified upper ocean, its full three-dimensional state remains only partially observable in the present experiments; a quantitative assessment of this limitation is left for future study."*

**Comment 9.**
Figure 8 shows how DA helps to recover the trajectory of a specific floe under sparse observational conditions. However, it seems that the floe under scrutiny is observed at every assimilation cycle within the period analyzed. If this is not the case, please indicate how many times the floe is observed and show this visually.

**Response:** We thank the reviewer for pointing this out. In the manuscript, the term "sparse observations" refers to the scenario in which approximately 30% of the floes are observed at each assimilation time, whereas "plentiful observations" corresponds to more than 70%. In both scenarios the assimilation cycle is 24.2 h. Floes that are not observed at a given cycle are not excluded; instead, their observation-error variance is inflated according to Eq. (50), thereby reducing their influence in the analysis. This is why at each assimilation cycle in Figure 8, the floe is always observed; however, the "observed" refers to floe location observed in small uncertainty and the "unobserved" refers to floe location observed in large uncertainty.

**Comment 10.**
Please add Normalized RMSE values for the prior, in addition to the posterior values, to better assess assimilation effectiveness.

**Response:** We thank the reviewer for this valuable suggestion. We followed the suggestion, we have added the Normalized Root Mean Square Error (RMSE) values in Tables 6-8 for the prior (background) state in addition to those for the posterior (analysis) state. This inclusion provides a more quantitative assessment for prior and posterior comparison.

**2 Minor Comments**

Line 181: there is a reference to the background vertical gradients of total water mixing ratio, but Table 4 reports a value for the background vertical gradients of total water. Is there a difference between total water and total water mixing ratio?

**Response.** We thank the reviewer for pointing out this clarification regarding the terminology and the background vertical gradients.

In this work, we use the term total water to refer to the total water mixing ratio, consistent with the formulation in [Edwards et al., 2020b, Edwards et al., 2020a]. Specifically, the background vertical gradient $d\tilde{q}_t/dz$ reported in Table 4 corresponds to the background gradient of the total water mixing ratio. This is also the quantity referenced in Line 181.

To avoid confusion, we have added one sentence in line 435 in the revised manuscript to state explicitly that total water and total water mixing ratio are used interchangeably in our presentation:

> *In addition, in this work, the terms "total water" and "total water mixing ratio" are used interchangeably, following the convention adopted in [Edwards et al., 2020a, Edwards et al., 2020b], to maintain consistency and clarity.*

Is the replacement of contact forces with white noise intended to increase ensemble spread in DA? Please clarify.

**Response.** Thank you for raising this point. The stochastic replacement of deterministic contact forces by a zero-mean white-noise term has two purposes, and only the second is related to ensemble spread:

- *Sub-grid representation of unresolved collisions.* In reality, floes experience numerous brief contacts whose individual forces are not resolved at the model grid spacing and time step used here. A white-noise approximation ($\sigma_f\, \eta(t)$ with $\langle\eta(t)\eta(t')\rangle = \delta(t - t')$) is a standard surrogate for the net impulse of many small, rapidly varying collisions. The amplitude $\sigma_f$ is tuned to match the observed mean-square horizontal velocity of floes in high-resolution simulations, ensuring that the noise has a physically interpretable magnitude.

- *Implicit model-error perturbation in the DA ensemble.* Because the same stochastic term is applied independently to each ensemble member, it also acts as a time-correlated model-error perturbation, naturally widening the spread of the prior without the need for explicit inflation. Thus the noise does increase ensemble spread, but that benefit is secondary; the primary motivation is to capture unresolved contact dynamics in a statistically consistent manner.

We have revised line 298 in Section 3.1.1:

> *"The stochastic forcing is used to effectively approximate the instantaneous floe-floe interactions in the forecast system, significantly reducing the computational cost while at the same time, providing a physically motivated source of model-error variance that maintains adequate ensemble spread. "*

Line 309: what does $\widetilde{\phi}_{bt,k}$ refer to?

**Response.** $\widetilde{\phi}_{bt,k}$ denotes the $k$-th Fourier coefficient of the barotropic component of the stream-function $\psi$; the subscript "$bt$" stands for "barotropic".

To make it clear, in the revised manuscript, we added the following in Section 3.1.2:

> "The subscript "bt" therefore labels the barotropic mode, while "bc" labels the baroclinic component."

Line 367: Equation (50) suggests larger floes have higher uncertainty, which contradicts earlier claims. Please clarify or correct the formula.

**Response.** Thank you for highlighting this potential confusion. Equation (50) prescribes the *standard deviation of observation errors* $\sigma_l^{\mathrm{obs}}$ for each floe according to the cloud–water criterion:

- **Observed floes** $[\ q_t < \widetilde{q}_t\ ]$ — The floe is under mostly clear sky and is therefore detected unambiguously; its positional uncertainty is set equal to the measurement uncertainty: $5 \times 10^2$ m. This error is *independent* of floe size.

- **Unobserved floes** $[\ q_t \geq \widetilde{q}_t\ ]$ — Thick cloud cover prevents a trustworthy observation of most of the area of the floe. Instead of discarding a cloud-obscured floe, we still include its location as an observation, but we recognize that this guess may be wrong by an amount comparable to the floe's own diameter.

$$\sigma_l^{\mathrm{obs}} = 2r_l,$$

This radius-scaled uncertainty represents the realistic possibility that the true floe could lie anywhere close the floe outline, providing a physically upper bound on positional error. Because the corresponding Kalman gain is nearly zero, this surrogate observation has minimal influence on the analysis; the $r_l$ scaling therefore does *not* imply that larger floes are inherently measured less accurately.

Thus, dependence on radius appears *only* for floes that are *not* observed; for any floe that is actually observed the uncertainty remains the same measurement uncertainty.

Line 385: There is again confusion between total water content and total water mixing ratio.

**Response.** We appreciate the reviewer's diligence in flagging this inconsistency. To avoid confusion, we have added one sentence in line 435 in the revised manuscript to state explicitly that total water and total water mixing ratio are used interchangeably in our presentation:

> *In addition, in this work, the terms "total water" and "total water mixing ratio" are used interchangeably, following the convention adopted in [Edwards et al., 2020a, Edwards et al., 2020b], to maintain consistency and clarity.*

Line 447: What is the difference between Equations (49) and (55)? They appear to be similar—clarification would help.

**Response.** We agree that Equations (49) and (55) are the same. Equation (55) was an inadvertent repetition introduced during a previous edit; it adds no new information beyond the definition already given in Eq. (49). To eliminate the redundancy and avoid confusion, we have deleted Eq. (55) and renumbered the subsequent equations accordingly.

Line 462: In Equation (50) the uncertainty below the threshold was set at 0.5 km.

**Response.** We thank the reviewer for identifying this typo. Line 462 has been corrected to read:

> *The observational uncertainty for the ice floe locations, when there is no cloud cover, is set to 0.5 km.*

Line 493: in Equation (50) is the threshold.

**Response.** We thank the reviewer for identifying this typo. Line 462 has been corrected to read:

> *The observational uncertainty, introduced via (50), shows that when cloud cover, $[q_t]$, remains below a certain threshold $\widetilde{q}_t$, the uncertainty stays small (around 500 m), allowing for accurate floe location recovery.*

Add a list of symbols with definitions and units of measurements.

We thank the reviewer for this suggestion. The symbols, definitions, and units for the variables used in the ice floe, atmospheric, and oceanic models are already provided in Tables 2–4 of the manuscript. These tables collectively serve as a reference for the key quantities associated with each model component. To improve clarity and ensure consistency in notation, we have added the following note at the beginning of Section 4.1:

> *The symbols with definitions and units for the ice floe model are listed in Table 2, for the ocean model in Table 3, and for the atmospheric model in Table 4.*

**References**

[Boisvert et al., 2023] Boisvert, L. N., Webster, M. A., Parker, C. L., and Forbes, R. M. (2023). Rainy days in the arctic. *Journal of Climate*, 36(19):6855–6878.

[Edwards et al., 2020a] Edwards, T. K., Smith, L. M., and Stechmann, S. N. (2020a). Atmospheric rivers and water fluxes in precipitating quasi-geostrophic turbulence. *Quarterly Journal of the Royal Meteorological Society*, 146(729):1960–1975.

[Edwards et al., 2020b] Edwards, T. K., Smith, L. M., and Stechmann, S. N. (2020b). Spectra of atmospheric water in precipitating quasi-geostrophic turbulence. *Geophysical & Astrophysical Fluid Dynamics*, 114(6):715–741.

[Qi and Majda, 2016] Qi, D. and Majda, A. J. (2016). Low-dimensional reduced-order models for statistical response and uncertainty quantification: Two-layer baroclinic turbulence. *Journal of the Atmospheric Sciences*, 73(12):4609–4639.

[Steele et al., 1997] Steele, M., Zhang, J., Rothrock, D., and Stern, H. (1997). The force balance of sea ice in a numerical model of the arctic ocean. *Journal of Geophysical Research: Oceans*, 102(C9):21061–21079.

[Thorndike and Colony, 1982] Thorndike, A. and Colony, R. (1982). Sea ice motion in response to geostrophic winds. *Journal of Geophysical Research: Oceans*, 87(C8):5845–5852.

[Thorndike, 1986] Thorndike, A. S. (1986). Kinematics of sea ice. In *The geophysics of sea ice*, pages 489–549. Springer.

---

## Author Comment (AC3)

**Response to Referee #3**

Changhong Mou, Samuel N Stechmann, Nan Chen*

[May 29, 2025]

**1    Comments**

- The subsections 2.2.2–2.2.4 should be one subsection, same for subsections 2.2.5–2.2.7. This would also match the authors intent reflected in Figure 1b about the two drag forces.

**Response:** We thank the reviewer for the helpful suggestion. We have combined Subsections 2.2.2–2.2.4 into one subsection titled "Oceanic forcing", and Subsections 2.2.5–2.2.7 into "Atmospheric forcing".

- In Section 2, the draft does not seem to mention what happens when the sea ice floes hit the boundary of the spatial domain. Boundary conditions for the atmosphere dynamics seem to be missing as well.

**Response:** We thank the reviewer for pointing this out. In our model, periodic boundary conditions are applied to the sea ice floes, as well as to the ocean and atmospheric components. This choice eliminates artificial boundary effects and reflects the spatial homogeneity assumed in our idealized setup.

To make it clear, we have revised our manuscript: (i) for the sea ice floes, we have added the following in line 101: "Periodic boundary conditions are imposed in both the $x$ and $y$ directions to ensure continuity in the domain." and (ii) for atmosphere dynamics, we have added the following description between line 165 and 170 to explicitly state that: "The two-layer fully saturated PQG model with periodic boundary conditions imposed in both the $x$ and $y$ directions yields the following equations."

- Both Brownian motions and Wiener process are used. Maybe stick to one of them?

**Response:** We appreciate the reviewer's pointing out the notation difference. To maintain consistency throughout the manuscript, we have revised the text in "line 306" to use the term Wiener process exclusively when referring to stochastic forcing.

- Both "the l-th floe" and "the lth floe" are used in the manuscript.

**Response:** We appreciate the reviewer's comment. We have revised the manuscript to enforce consistent notation "l-th floe" throughout the draft.

- Page 10, Line 194, use $\Phi_1^a$ and $\Phi_2^a$ instead of $\Phi_1$ and $\Phi_2$?

**Response:** We appreciate the reviewer's comment. We have revised the manuscript to correct the notation.

- There is a discrepancy between the $\Delta t$ value reported in lines 399, 459 and in Table 2.

**Response:** We thank the reviewer for carefully identifying this inconsistency. To clarify, the atmospheric model uses a time step of $\Delta t^{\mathrm{a}} = 58.2$ seconds, while the discrete element method (DEM) sea ice model, to reduce the computational cost, operates on a coarser time step of $\Delta t^{\mathrm{ice}} = 80 \times \Delta t^{\mathrm{a}} \approx 1.29$ hours. In the data assimilation setup, observations are assumed to be available every 1500 atmospheric model time steps, corresponding to approximately $\Delta t^{\mathrm{obs}} = 24.2$ hours which is close to the true frequency of acquiring satellite images.

We haved added the following in the revised manuscript before Table 5.:

> " It is worth noting that the atmospheric model uses a time step of $\Delta t = 58.2$ seconds, while the DEM sea ice model is updated every 80 atmospheric steps, giving $\Delta t \approx 1.29$ hours. Observations for data assimilation are assumed to be available every 1500 atmospheric steps, or $\Delta t^{obs} \approx 24.2$ hours."

- Please check the physical units for $E_i$ and $E_o$ in Table 2.

**Response:** When we look at the units of $E_i$ and $E_o$ again, we can see why it may have caught your attention. We are not using the units of a surface flux, which typically include $\mathrm{m}^{-2}$ to describe the flux of a quantity across a unit area. Instead, we are using units of $\mathrm{kg\,kg}^{-1}\,\mathrm{s}^{-1}$, which represent the change of the atmospheric total water mixing ratio per unit time. We use these units, rather than the units of a flux, because vertical transport is not resolved in the atmospheric boundary layer in our model. Instead, evaporation is represented in a simple parameterized form, and it represents the change in (height-averaged) total water mixing ratio in the lower tropospheric layer.

- Please check the physical units for the background vertical gradient of total water in Table 4.

**Response:** We have double checked the units of $d\tilde{q}_t/dz$, and we think they are correct in the original submission as $\mathrm{kg\,kg}^{-1}\,\mathrm{km}^{-1}$, which represents the change in the mixing ratio (units of

kg kg$^{-1}$) per change in height. We have also double checked that this is also used in the reference [Edwards et al., 2020] (aside from different choices in units of mixing ratio of g kg$^{-1}$ versus kg kg$^{-1}$).

**References**

[Edwards et al., 2020] Edwards, T. K., Smith, L. M., and Stechmann, S. N. (2020). Atmospheric rivers and water fluxes in precipitating quasi-geostrophic turbulence. *Quarterly Journal of the Royal Meteorological Society*, 146(729):1960–1975.

---

## Referee Report (RR1)

I thank the authors for their comprehensive responses to my questions and comments. Most of the proposed revisions are appropriate and well-integrated.
However, I would appreciate further clarifications regarding the treatment of ice floes observations.

- In the response to comment 9, the authors state that "*Floes that are not observed at a given cycle are not excluded; instead, their observation-error variance is inflated according to Eq. (50), thereby reducing their influence in the analysis*". This seems to confict with Eq. (54) which sets out a different criterion, based on the local total water content and specifies conditions under which a floe cannot be observed at all. Moreover, if "not observed" refers to an inflated observation error (as per Eq. 50), it appears that the same equation also determines the observation level (plentiful or sparse), based on the mean total water content [$q_t(\mathbf{x},t)$].  I would appreciate clarification on how observational availability and uncertainty are operationally determined.
In this context, I think that it would be beneficial to merge paragraphs 3.2 and 3.3.1 and to make a clear distinction between true floes' coordinates and observations, adding an equation that links the two.

- An additional comment regarding Eq. 50: is there a defined lower bound on the observational uncertainty in the case of significant cloud cover? According to the current formulation, it seems possible that a small floe with high mean total water content could yield a lower observational uncertainty than a floe under clear conditions. This seems counterintuitive and may require a justification.

**Minor points:**

- Line 309 (line 345 in the revised ms) : Apologies for not being clear earlier — I was referring to the tilde accent, which appears to be a typographical error.

---

## Author Response (AR2)

**Response to Referee #2**

Changhong Mou, Samuel N Stechmann, Nan Chen*

[June 23, 2025]

We thank the reviewer for the positive feedback and are glad that most of our revisions were well received. Below we provide a point-by-point response to the remaining comment.We hope this revision addresses the reviewer's concern.

**1   General Comments**

**Comment 1.**
In the response to comment 9, the authors state that "Floes that are not observed at a given cycle are not excluded; instead, their observation-error variance is inflated according to Eq. (50), thereby reducing their influence in the analysis". This seems to conflict with Eq. (54) which sets out a different criterion, based on the local total water content and specifies conditions under which a floe cannot be observed at all. Moreover, if "not observed" refers to an inflated observation error (as per Eq. 50), it appears that the same equation also determines the observation level (plentiful or sparse), based on the mean total water content [qt(x,t)]. I would appreciate clarification on how observational availability and uncertainty are operationally determined. In this context, I think that it would be beneficial to merge paragraphs 3.2 and 3.3.1 and to make a clear distinction between true floes' coordinates and observations, adding an equation that links the two.

**Response:** We thank the reviewer for identifying the potential ambiguity and for the helpful structural suggestion. We agree that confusion may arise. Indeed, in Eq.(54)), "not observed" denotes a floe whose observation–error variance is inflated according to Eq.(50)) and Adjusting the threshold in Eq.(50)) therefore changes the observation level. Here, Eq (54). also reminds readers that, in the data assimilation, LETKF scheme, a floe is treated as unobserved and therefore assigned large uncertainty—whenever $q_t$ exceeds this threshold. We follow the reviewer's suggestion to merge paragraphs related to observability in Sec 3.3.1 to Sec 3.2. The revised part in Sec 3.2 yields the following:

*To represent observational uncertainty in DA, we use the total water content $q_t(\mathbf{x}, t)$ as a controlling factor. Above each floe, we calculate the mean total water content, $[q_t(\mathbf{x}, t)]$ at time $t$:*

$$[q_t(\mathbf{x}, t)] = \frac{1}{|\Omega_l|} \int_{\Omega_l} q_t(\mathbf{x}, t) d\mathbf{x}. \tag{1}$$

*This mean value, $[q_t(\mathbf{x}, t)]$, encapsulates the spatial distribution of water content above each ice floe, serving as an approximation for the uncertainty in observations. Variations in $[q_t(\mathbf{x}, t)]$ from one floe to another can indicate the degree of uncertainty inherent*

*in the observational data, as it reflects the heterogeneity in the physical characteristics of the ice. In particular, we set a threshold, $\widetilde{q}_t$, such that the observational uncertainty $\sigma_l^{obs}$ is given by the following: for l-th floe,*

$$\sigma_l^{\mathrm{obs}}\big(\mathbf{x}_l(t)\big) = \begin{cases} 5 \times 10^2 \text{ m}, & \text{if } [q_t(\mathbf{x},t)] < \widetilde{q}_t \ (small\ observation\ uncertainty), \\ 2r_l, & \text{if } [q_t(\mathbf{x},t)] \geq \widetilde{q}_t \ (large\ observation\ uncertainty). \end{cases} \tag{2}$$

*where $r_l$ denotes the radius of the l-th floe and $\mathbf{x}_l$ its trajectory. It is important to note that when the mean total water content over the floe, $[q_t(\mathbf{x},t)]$, is high, which indicates significant cloud cover and can be classified as* unobserved*, its position can still be approximated. However, these estimates are often highly inaccurate. Consequently, in the data-assimilation setting, we assign floes classified as* unobserved *a markedly inflated observational uncertainty, taken here as twice their radius.*

**Comment 2.**
An additional comment regarding Eq. 50: is there a defined lower bound on the observational uncertainty in the case of significant cloud cover? According to the current formulation, it seems possible that a small floe with high mean total water content could yield a lower observational uncertainty than a floe under clear conditions. This seems counterintuitive and may require a justification.

**Response:** We thank the reviewer for raising this question. In our scheme a floe classified *unobserved* (i.e. $[q_t] \geq \widetilde{q}_t$) is always assigned
$$\sigma_l^{\mathrm{obs}} = 2r_l.$$
Because this uncertainty is on the order of the floe's diameter—even for the smallest floes—the observation contributes negligibly to the analysis. Also, it it is worthwhile to note that in all test cases the minimum floe radius is $8 \times 10^3$ m; hence a floe beneath thick cloud is assigned at least $2r_l = 1.6 \times 10^4$ m of uncertainty, far exceeding lower bound of uncertainty when the floe is under clear sky of $5 \times 10^2$ m.

To clarify this point for readers, we have added the following remark in the end of Sec. 3.2. in revised manuscript:

> ***Remark.*** *An ice floe is classified as "unobserved" whenever the mean total water content over the floe exceeds the threshold, $[q_t] \geq \widetilde{q}_t$ (cf. Eq. (50)). In that case we inflate the observation–error standard deviation to $\sigma_l^{\mathrm{obs}} = 2r_l$, with $r_l$ the floe radius. Because this value is of the same order as the floe's diameter, the associated observation exerts negligible influence on the analysis. Among all test cases the smallest floe radius is $r_l^{\min} = 8 \times 10^3$ m, so an unobserved floe takes at least $2r_l = 1.6 \times 10^4$ m of uncertainty, clearly distinguished from the observed case with $\sigma_l^{obs} = 5 \times 10^2$ m.*

**2  Minor Points**

**Comment 1.**
- Line 309 (line 345 in the revised ms) : Apologies for not being clear earlier — I was referring to the tilde accent, which appears to be a typographical error.

**Response:** We thank the reviewer for pointing this out and apology for this typo. We have revised the manuscript and correct it to $\widehat{\psi}_{bt,\mathbf{k}}$.